# Assessing Uncertainty in Similarity Scoring: Performance & Fairness in Face Recognition

**Jean-Rémy Conti**[1,2] **& Stéphan Clémençon**[1]
[1]LTCI, Télécom Paris, Institut Polytechnique de Paris, [2]Idemia
{jean-remy.conti,stephan.clemencon}@telecom-paris.fr *

## Abstract

The ROC curve is the major tool for assessing not only the performance but also the fairness properties of a similarity scoring function. In order to draw reliable conclusions based on empirical ROC analysis, accurately evaluating the uncertainty level related to statistical versions of the ROC curves of interest is absolutely necessary, especially for applications with considerable societal impact such as Face Recognition. In this article, we prove asymptotic guarantees for empirical ROC curves of similarity functions as well as for by-product metrics useful to assess fairness. We also explain that, because the false acceptance/rejection rates are of the form of U-statistics in the case of similarity scoring, the naive bootstrap approach may jeopardize the assessment procedure. A dedicated recentering technique must be used instead. Beyond the theoretical analysis carried out, various experiments using real face image datasets provide strong empirical evidence of the practical relevance of the methods promoted here, when applied to several ROC-based measures such as popular fairness metrics.

## 1 Introduction

The massive deployment of AI technologies brings with it a pressing demand for methodological tools to assess their trustworthiness. The reliability of AI systems concerns their estimated performance of course, but also their properties regarding fairness: ideally, the system should exhibit approximately the same performance, independently of the *sensitive* group (determined by *e.g.* gender, age group, race) to which it is applied. This is particularly true for Face Recognition (FR) systems, the running example through this article, now under scrutiny by the general public and regulatory organizations, refer to *e.g.* Grother & Ngan (2019) or Snow (2018).

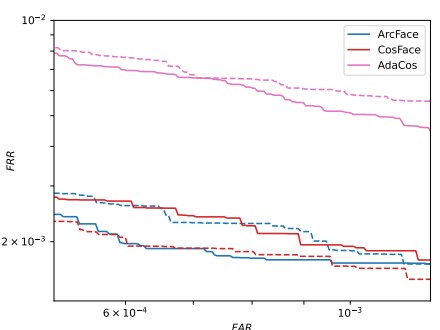

Figure 1: Empirical ROC curves for three different models (ArcFace, CosFace, Ada-Cos) and for two distinct evaluation datasets (see 4). The ROC curves for the first dataset are depicted with solid lines while the ROC curves for the second dataset are displayed with dashed lines. A confidence band for the ROC computed with the ArcFace model on the first dataset is displayed in light blue.

The task of designing a FR system is usually formulated as a similarity scoring/learning problem, see *e.g.* Vogel et al. (2018). Assuming that the system processes pixellated face images $X$ in $\mathbb{R}^d$, the goal is to build a (symmetric) scoring function $s : \mathbb{R}^d \times \mathbb{R}^d \to \mathbb{R}$ such that the larger the similarity score $s(x, x')$ related to a pair of images, the larger (hopefully) the probability that both images correspond to the same individual. In this case, one assigns a positive label to the pair and a negative label otherwise. The gold standard to measure the performance of such a similarity $s$ is the Receiver Operating Characteristic (ROC) curve (D.M.Green & Swets, 1966), namely the plot of the false rejection rate against the false acceptance rate, as the similarity scoring threshold varies.

---

*Alternative correspondence: jeanremy.conti@gmail.com.

If, until now, the benchmark of FR systems has been essentially reduced to an ad-hoc evaluation of the performance metrics, to the computation of empirical ROC curves based on FR evaluation datasets of reference (see Grother & Ngan (2019)), the quantification of the uncertainty inherent in the randomness of the evaluation datasets (referred to as *aleatoric uncertainty* sometimes, see Hüllermeier & Waegeman (2021)) is essential to compare and appreciate fully the merits of such systems regarding accuracy and fairness with confidence. The significance of the uncertainty quantification step is illustrated in Fig. 1. The ROC curves for three different FR models are computed on two distinct evaluation datasets (see 4 for details on models and datasets). On the first dataset (solid lines), one would conclude that ArcFace is a better model than CosFace, as the former has a lower empirical FRR than the latter, for any FAR value. However, one would draw the opposite conclusion, looking at the second dataset (dashed lines). Selecting models only depending on their empirical performance (ROC) has its flaws, as the uncertainty of those metrics is not taken into account. Note that the method for building confidence bands for the ROC, which we present in this paper, would have avoided such conclusions. Indeed, both models are indistinguishable in terms of performance, as their ROC curves are contained within the band. On the contrary, the AdaCos model performs worse than ArcFace/CosFace on both datasets and its empirical ROC curves are far from the confidence band. One could thus favor confidently ArcFace/CosFace over AdaCos regarding their performance.

In order to make meaningful comparisons, the (possibly high) uncertainty inherent in the statistical nature of the estimation must be taken into account. Indeed, this evaluation is crucial to judge whether the similarity scoring function candidates meet the performance/fairness requirements in a trustworthy manner. The main purpose of this paper is precisely to explain how to quantify the uncertainty/variability of similarity ROC curves, by means of a dedicated bootstrap methodology in particular, in a sound validity framework generalizing the one established by Bertail et al. (2008) in the non pairwise setup.

**Related works.** To our knowledge, the uncertainty issue inherent to ROC curve estimation and fairness metrics estimation is poorly documented in the literature, particularly for similarity scoring problems such as FR. It has been studied at length for scoring functions, but in the non pairwise setup, using bootstrap methods in (Bertail et al., 2008). The major difference between the analysis carried out therein and our framework lies in the fact that, in the similarity scoring context, false acceptance/rejection rates are not basic i.i.d. averages anymore but generalized $U$-statistics (refer to Lee (1990) or Arcones & Gine (1992)). It has a significant impact on the methodology that can be used to quantify the uncertainty of empirical performance/fairness measures. As shall be seen in this paper, naively applying the bootstrap of Bertail et al. (2008) to similarity scoring problems (*e.g.* FR) strongly underestimates the ROC curve, resulting in confidence bands for the ROC curve which do not even contain the empirical ROC curve most of the time. In Vogel et al. (2018), non asymptotic confidence bounds for the estimation error of empirical similarity ROC curves have been established by means of linearization techniques tailored to $U$-statistics in a slightly different probabilistic framework (stipulating random labels). It is the purpose of the present paper to investigate how to accurately approximate the distribution of the estimation error by means of dedicated resampling techniques and build bootstrap confidence bands with satisfactory probability coverage.

**Contributions.** As will be explained in the subsequent analysis, a naive application of the bootstrap procedure yields a systematic underestimation of the similarity ROC curve. We provide *(i)* a recentering technique to counteract this, while still being accurate asymptotically. Resulting from this bootstrap variant, *(ii)* confidence bands for the ROC curve and FR fairness metrics are shown to be consistent, in addition to achieve nominal coverage on synthetic data. The recentered bootstrap also allows to define *(iii)* a scalar uncertainty measure for the ROC and fairness metrics, which can be employed to compare the robustness of several FR fairness metrics. Finally, in addition to the statistical analysis presented, the relevance of the approach is supported by *(iv)* illustrative numerical experiments, based on real data of face images, together with a discussion about the practical use of the information produced, in order to make more reliable decisions concerning accuracy and fairness. These results pave the way for a more valuable and trustworthy comparative analysis of the merits and drawbacks of FR systems.

**Organization of the paper.** The main concepts at work in similarity scoring/learning and in FR are briefly recalled in Section 2, together with the notions pertaining to ROC analysis used in this article to evaluate predictive performance and build fairness criteria. The consistency of empirical similarity ROC curves, is stated in section 3. It is also explained therein how to bootstrap empirical similarity ROC curves in a valid manner, as well as by-product summary statistics reflecting the

accuracy or fairness properties of the similarity scoring functions under study. Numerical experiments are presented and discussed in section 4 for illustration purpose.

## 2 BACKGROUND AND PRELIMINARIES

We introduce here the main notations used throughout the article and briefly recall the standard similarity scoring/learning framework, involved in the design of FR systems in particular, and the key concepts pertaining to ROC analysis that are involved in the subsequent study. We next explain how to formulate fairness criteria based on similarity ROC curves in the FR context. Here and throughout, the indicator function of any event $\mathcal{E}$ is denoted by $\mathbb{I}\{\mathcal{E}\}$, the Dirac mass at any point $x$ by $\delta_x$, and the pseudo-inverse of any cumulative distribution function (cdf) $\kappa(t)$ on $\mathbb{R}$ by $\kappa^{-1}(\alpha) = \inf\{t \in \mathbb{R} : \kappa(t) \geq \alpha\}$.

### 2.1 SIMILARITY SCORING - PROBABILISTIC AND STATISTICAL FRAMEWORK

The probabilistic framework considered here to formulate the similarity learning problem is the same as that of multi-class classification: $Y$ is a discrete random label defined on a probability space $(\Omega, \mathcal{A}, \mathbb{P})$, valued in $\mathcal{Y} = \{1, \ldots, K\}$ with $K \geq 2$, and $X$ is a random vector defined on the same probability space and taking its values in a high dimensional space $\mathcal{X} \subset \mathbb{R}^d$ with $d \gg 1$. For all $k \in \mathcal{Y}$, we denote the supposedly continuous conditional distribution of $X$ given $Y = k$ by $F_k$, the probability that $Y$ equals $k$ by $p_k = \mathbb{P}\{Y = k\}$. Equipped with these notations, the joint distribution $P$ of the random pair $(X, Y)$ is fully characterized by $\{(F_k, p_k) : k = 1, \ldots, K\}$. In the running example considered through this paper, $X$ is an image depicting the face of an individual in a population of $K$ identities, the identity being indexed by $Y$. In similarity learning, the objective pursued is to find a mapping $s : \mathcal{X}^2 \to \mathbb{R} \cup \{+\infty\}$, called a *similarity scoring function*, such that, given two independent pairs $(X, Y)$ and $(X', Y)$, the larger the similarity score $s(X, X')$, the more likely the same label should be shared (*i.e.* one should observe the event $Y = Y'$) ideally. Before recalling performance/fairness metrics in similarity scoring, we explain the usual methodology at work in FR.

**Similarity scoring in Face Recognition.** In FR, one learns, from a dataset of face images with identity labels, an encoder function $f : \mathbb{R}^{h \times w \times c} \to \mathbb{R}^p$ that embeds the images in a way that brings same identities closer together in a certain sense. Each image is of size $(h, w)$, while $c$ corresponds to the color channel dimension. It is worth noticing that a pre-processing detection step (finding a face within an image) is required so that all face images have the same size $(h, w)$. For an image $x \in \mathbb{R}^{h \times w \times c}$, its latent representation $f(x) \in \mathbb{R}^p$ is referred to as the face embedding of $x$. Since the advent of deep learning, the encoder $f$ is usually a deep Convolutional Neural Network (CNN) whose parameters are learned on a huge FR dataset, made of face images and identity labels. In brief, the training consists in taking all images $x_i^{(k)}$, labelled with identity $k$, computing their embeddings $f(x_i^{(k)})$ and adjusting the parameters of $f$ so that those embeddings are as close as possible (for a given similarity measure) and as far as possible from the embeddings of identity $l \neq k$. The usual similarity measure is the *cosine similarity*, defined as

$$s(x_i, x_j) := \frac{f(x_i)^\mathsf{T} f(x_j)}{\|f(x_i)\| \cdot \|f(x_j)\|} \tag{1}$$

for two images $x_i$ and $x_j$, where $\|\cdot\|$ stands for the usual Euclidean norm. In some early works (Schroff et al., 2015), the Euclidean metric $\|f(x_i) - f(x_j)\|$ is also used.

**Multi-sample statistical setup.** In the subsequent analysis, the $p_k$'s are supposed to be given and are part and parcel of the performance criterion (a possible choice consists in giving the same weight to all identities, *i.e.* $p_k = 1/K$ for $k = 1, \ldots, K$), whereas the $F_k$'s are unknown in practice. In order to evaluate empirically the properties of a (trained) similarity scoring function $s(x, x')$, it is assumed throughout the paper that $K$ i.i.d. samples are available:

$$X_1^{(k)}, \ldots, X_{n_k}^{(k)} \overset{i.i.d.}{\sim} F_k, \text{ where } n_k \geq 1 \text{ for } k = 1, \ldots, K.$$

In the FR context, we thus suppose that $n_k$ images are available for identity $k \in \{1, \ldots, K\}$, the size of the pooled sample being denoted by $n = n_1 + \ldots + n_K$. Those images belong to a test dataset, used to evaluate a trained FR model $f$ (or equivalently its associated similarity $s$ in Eq. 1). Based on

these data, empirical versions of performance and fairness metrics can be computed. Their statistical accuracy will be next investigated from an asymptotic perspective, as the $n_k$'s simultaneously tend to infinity at the same rate as $n$: for all $k \in \mathcal{Y}$, there exists $\lambda_k > 0$ such that $n_k/n \to \lambda_k$ as $n \to +\infty$.

## 2.2 ROC ANALYSIS - EVALUATION OF PERFORMANCE/FAIRNESS IN SIMILARITY SCORING

As formulated in Vogel et al. (2018), similarity learning can be seen as a specific *bipartite ranking* problem, where the input space is of the form of a product space $\mathcal{X} \times \mathcal{X}$: given two independent observations $(X, Y)$ and $(X', Y')$ drawn from $\mathbb{P}$, the input r.v. is formed by the pair $(X, X')$, while $Z = 2\mathbb{I}\{Y = Y'\} - 1$ is the binary label. The gold standard to evaluate bipartite ranking performance is ROC analysis: a statistical learning view can be found in *e.g.* Clémençon & Vayatis (2009). In the similarity learning context, the ROC curve of a similarity scoring function $s(x, x')$ is the plot of the False Rejection Rate (FRR) against the False Acceptance Rate (FAR) as the acceptance threshold varies, namely the mapping $\mathrm{ROC}\colon \alpha \in (0, 1) \mapsto \mathrm{FRR} \circ \mathrm{FAR}^{-1}(\alpha)$, where, for all $t \in \mathbb{R}$,

$$
\begin{aligned}
\mathrm{FAR}(t) &= \mathbb{P}\{s(X, X') > t \mid Z = -1\} = \frac{\sum_{k<l} p_k p_l \mathbb{P}\{s(X, X') > t \mid Y = k,\ Y' = l\}}{\sum_{k<l} p_k p_l}, \\
\mathrm{FRR}(t) &= \mathbb{P}\{s(X, X') \le t \mid Z = +1\} = \frac{\sum_{k\in\mathcal{Y}} p_k^2 \mathbb{P}\{s(X, X') \le t \mid Y = Y' = k\}}{\sum_{k\in\mathcal{Y}} p_k^2}.
\end{aligned}
$$

A note on the definition of the pseudo-inverse for the FAR quantity is available in D.1.

**Remark 1.** (ROC CONVENTIONS) *In machine learning, the* ROC *curve usually refers to the PP-plot $t \in \mathbb{R} \mapsto (\mathrm{FAR}(t), 1 - \mathrm{FRR}(t))$, or equivalently $\alpha \in (0, 1) \mapsto 1 - \mathrm{FRR} \circ \mathrm{FAR}^{-1}(\alpha)$. The FR community preferably plots $\mathrm{FAR}(t)$ on the $x$-axis and $\mathrm{FRR}(t)$ on the $y$-axis. Both components correspond to error rates that should be minimized, one possibly more than the other depending on the use case. Of course, these two curves provide exactly the same information as there is a one-to-one correspondence between them. Note that we use the FR convention throughout the paper.*

In practice, special attention is paid to certain points of the ROC curve. The FAR level $\alpha \in (0, 1)$ determines the operational point of the FR system and corresponds to the security risk one is ready to take. According to the FR use case, it is typically set to $10^{-i}$ with $i \in \{1, \dots, 9\}$.

**Fairness metrics.** In order to inspect the fairness properties of a FR system based on a similarity scoring function $s$, one generally looks at differentials in performance amongst several subgroups/segments of the population, a *sensitive attribute* (*e.g.* gender, race, age class, ...) making them distinguishable. For a given (discrete) sensitive attribute that can take $M > 1$ different values, in $\mathcal{A} = \{0, 1, \dots, M - 1\}$ say, we enrich the probability space and now consider a random vector $(X, Y, A)$ where $A \in \mathcal{A}$ indicates the subgroup to which the individual indexed by $Y$ belongs to. For every fixed value $a \in \mathcal{A}$, we can further define the FAR/FRR related to subgroup $a$: $\mathrm{FAR}_a(t) = \mathbb{P}\{s(X, X') > t \mid Y \ne Y',\ A = A' = a\}$ and $\mathrm{FRR}_a(t) = \mathbb{P}\{s(X, X') \le t \mid Y = Y',\ A = A' = a\}$ for all $t \in \mathbb{R}$, where by $(X', Y', A')$ is meant an independent copy of the random triplet $(X, Y, A)$. Ideally, a fair scoring function $s$ would exhibit nearly constant $\mathrm{FAR}_a(t)$ values when $a$ varies, for all $t$ (and the same property for the $\mathrm{FRR}_a(t)$ values). A FR fairness metric quantifies how much a model $s$ is far from this property. The FR fairness metrics considered in this paper are those used by the U.S. National Institute of Standards and Technology (NIST) in their FRVT report (Grother, 2022). They attempt to quantify the differentials in $(\mathrm{FAR}_a(t))_{a\in\mathcal{A}}$ and $(\mathrm{FRR}_a(t))_{a\in\mathcal{A}}$. Each fairness metric has two versions (one for the differentials in terms of FAR, the other in terms of FRR). A typical fairness metric is the max-min fairness below:

$$
\mathrm{FAR}_{\min}^{\max}(t) = \frac{\max_{a\in\mathcal{A}} \mathrm{FAR}_a(t)}{\min_{a\in\mathcal{A}} \mathrm{FAR}_a(t)}, \qquad \mathrm{FRR}_{\min}^{\max}(t) = \frac{\max_{a\in\mathcal{A}} \mathrm{FRR}_a(t)}{\min_{a\in\mathcal{A}} \mathrm{FRR}_a(t)}.
$$

In practice, the threshold $t$ is set as for the ROC curve, *i.e.* it achieves a level $\mathrm{FAR}(t) = \alpha \in (0, 1)$ for the global/total population, and not for some specific subgroup. Three other popular FR fairness metrics are the max-geomean metric, the log-geomean metric and the Gini coefficient. Their definition is postponed to A.1 for conciseness.

## 3    SIMILARITY SCORING METRICS - ASSESSING UNCERTAINTY

Motivated by the need to make trustworthy decisions taking into account the uncertainty inherent in the evaluation data, we now investigate the statistical accuracy of empirical counterparts of the ROC curve of a given similarity scoring function $s(x, x')$ in the statistical multi-sample framework described in 2.1. We next explain how to use the bootstrap methodology to estimate the related uncertainty level and build accurate confidence bands for the ROC and fairness metrics. For notational simplicity, the results are stated and proved in the case where $p_k = 1/K$ for all $k \in \mathcal{Y}$, extension to the general case being straightforward.

### 3.1    STATISTICAL INFERENCE - CONSISTENCY RESULT

An estimator of the ROC curve is naturally obtained by replacing the quantities FAR and FRR with their natural statistical counterparts in the definition of the ROC curve. In the multi-sample statistical setup defined in 2.1, the empirical versions of $\mathrm{FAR}(t)$ and $\mathrm{FRR}(t)$ can be expressed as follows, using the symmetry property of similarity scoring functions: for all $t \in \mathbb{R}$,

$$\widehat{\mathrm{FAR}}_n(t) = \frac{2}{K(K-1)} \sum_{k<l} \frac{1}{n_k n_l} \sum_{i=1}^{n_k} \sum_{j=1}^{n_l} \mathbb{I}\{s(X_i^{(k)}, X_j^{(l)}) > t\}, \tag{2}$$

$$\widehat{\mathrm{FRR}}_n(t) = \frac{1}{K} \sum_{k=1}^{K} \frac{2}{n_k(n_k-1)} \sum_{1 \le i < j \le n_k} \mathbb{I}\{s(X_i^{(k)}, X_j^{(k)}) \le t\}. \tag{3}$$

Notice that the terms involved in the two averages above are not independent, as each $X_i^{(k)}$ is involved in many terms of both averages, in contrast to the standard bipartite ranking framework (Bertail et al., 2008) where one deals with i.i.d. mean statistics. As detailed in A.2, these averages are actually generalized $U$-statistics, the simplest extensions of standard i.i.d. mean statistics. Properties and asymptotic theory of $U$-statistics can be found in Lee (1990) while concentration properties are investigated in Clémençon et al. (2016). The quantities (2) and (3) can then be used to compute the *empirical similarity* ROC *curve* based on the available evaluation datasets:

$$\widehat{\mathrm{ROC}}_n : \alpha \in (0, 1) \mapsto \widehat{\mathrm{FRR}}_n \circ (\widehat{\mathrm{FAR}}_n)^{-1}(\alpha). \tag{4}$$

Empirical versions of fairness metrics are naturally obtained in a similar *plug-in* fashion (see D.3).

The result stated below reveals the uniform consistency of the curve (4) in the multi-sample asymptotic framework considered here.

**Proposition 1.** (STRONG CONSISTENCY) *With probability one, we have:*

$$\sup_{\alpha \in (0,1)} \{\widehat{\mathrm{ROC}}_n(\alpha) - \mathrm{ROC}(\alpha)\} \to 0, \text{ as } n \to +\infty. \tag{5}$$

Refer to D.2 for the technical proof. While the true ROC curve is unknown, this result gives confidence in the quantity $\widehat{\mathrm{ROC}}_n$ one computes based on data. A similar consistency result is stated in Hsieh & Turnbull (1996), when the negative and positive cdf's are estimated by basic i.i.d. averages, and proved by means of classic results for empirical processes. The case of empirical similarity ROC curves cannot be handled in the same way because, as previously emphasized, (2) and (3) are generalized $U$-statistics and the terms involved in these averages exhibit a complex dependence structure. Linearization tricks (*i.e.* Hoeffding decomposition), such as those used in Vogel et al. (2018), would be required to establish in addition the asymptotic Gaussianity of (a rescaled version of) the fluctuation process

$$r_n(\alpha) := \sqrt{n}\{\widehat{\mathrm{ROC}}_n(\alpha) - \mathrm{ROC}(\alpha)\}, \quad \alpha \in (0, 1). \tag{6}$$

As underlined in Bertail et al. (2008), where the (much simpler) statistical framework considered is the same as in Hsieh & Turnbull (1996), the identification of the Gaussian limit of the process of Eq. 6 is of very poor interest regarding the construction of (asymptotic) confidence bands for the similarity ROC curve: beyond the computational difficulties inherent in simulating Brownian bridges, the presence of the unknown quantity $\mathrm{ROC}(t)$ in the complex limit law makes its use impracticable to build confidence bands. Appropriate bootstrap techniques, whose asymptotic validity can be proved, should be preferably used instead.

### 3.2 BOOTSTRAPPING THE PERFORMANCE/FAIRNESS METRICS - CONFIDENCE REGIONS

Provided that representative datasets of the target populations are available, the empirical ROC curve (4) (and its related scalar summaries) of a similarity scoring function $s(x, x')$ is the main tool to assess performance and fairness in various applications such as FR. However, in order to make meaningful comparisons, the (possibly high) uncertainty inherent in the statistical nature of the estimation must be taken into account. Indeed, this evaluation is crucial to judge whether the similarity scoring function candidates meet the performance/fairness requirements in a trustworthy manner, as will be discussed on real examples in the next section. We now explain how to use a specific bootstrap resampling methodology to quantify the variability of the fluctuation process (6) in an asymptotically valid manner and why a naive bootstrap technique fails in the present situation.

**Objective.** When computing $\widehat{\mathrm{ROC}}_n(\alpha)$ to estimate the true ROC curve, one makes the error

$$\hat{\epsilon}_n(\alpha) = \widehat{\mathrm{ROC}}_n(\alpha) - \mathrm{ROC}(\alpha), \tag{7}$$

which is unknown, just like $\mathrm{ROC}(\alpha)$. The variability of the random variable $\hat{\epsilon}_n(\alpha)$ fully characterizes the uncertainty of the empirical ROC curve. The objective is to approximate $\hat{\epsilon}_n(\alpha)$ so that its variability can be estimated. This variability (*i.e.* the uncertainty of $\widehat{\mathrm{ROC}}_n(\alpha)$) will be used to build confidence bands around the empirical ROC curve and to define a scalar uncertainty metric. In order to approximate $\hat{\epsilon}_n(\alpha)$, the bootstrap approach makes it possible to sample an estimate of $\hat{\epsilon}_n(\alpha)$. With many samples, one can retrieve the variability of the error $\hat{\epsilon}_n(\alpha)$.

**Naive bootstrap.** The bootstrap paradigm, introduced by Efron (1979) and developed at length in Bertail et al. (2008), suggests to recompute the empirical similarity ROC curve (4) from $K$ independent sequences of i.i.d. variables $X_1^{(k)*}, \ldots, X_{n_k}^{(k)*} \overset{i.i.d.}{\sim} \hat{F}_k = \frac{1}{n_k} \sum_{1 \le i \le n_k} \delta_{X_i^{(k)}}$, conditioned upon the evaluation dataset $\mathcal{D} := \{X_1^{(k)}, \ldots, X_{n_k}^{(k)} : k = 1, \ldots, K\}$. In other words, for each identity $k$ within the dataset, one simply randomly samples with replacement $n_k$ images $X_1^{(k)*}, \ldots, X_{n_k}^{(k)*}$ among the $n_k$ face images $X_1^{(k)}, \ldots, X_{n_k}^{(k)}$ available for identity $k$. The *bootstrap sample*, which should be viewed as a sort of replicate of the evaluation dataset, is obtained by concatenating all the images thus sampled. Notice that the bootstrap sample may contain the same images several times due to the use of the sampling with replacement scheme, in contrast to the original dataset $\mathcal{D}$. In practice, the resampling scheme is replicated $B \ge 1$ times in order to compute a Monte-Carlo approximation of the distribution of the *bootstrap* ROC, *i.e.* the curve $\alpha \mapsto \widehat{\mathrm{ROC}}_n^*(\alpha) := \widehat{\mathrm{FRR}}_n^* \circ (\widehat{\mathrm{FAR}}_n^*)^{-1}(\alpha)$, with

$$\widehat{\mathrm{FAR}}_n^*(t) = \frac{2}{K(K-1)} \sum_{k<l} \frac{1}{n_k n_l} \sum_{i=1}^{n_k} \sum_{j=1}^{n_l} \mathbb{I}\{s(X_i^{(k)*}, X_j^{(l)*}) > t\}, \tag{8}$$

$$\widehat{\mathrm{FRR}}_n^*(t) = \frac{1}{K} \sum_{k=1}^{K} \frac{2}{n_k(n_k-1)} \sum_{1 \le i < j \le n_k} \mathbb{I}\{s(X_i^{(k)*}, X_j^{(k)*}) \le t\}. \tag{9}$$

The curve $\widehat{\mathrm{ROC}}_n^*$ is nothing but the empirical ROC curve computed with a bootstrap sample, instead of the original dataset. It turns out that, conditionally to the dataset $\mathcal{D}$, the quantity $\hat{\epsilon}_n^{(1)}(\alpha) = \widehat{\mathrm{ROC}}_n^*(\alpha) - \widehat{\mathrm{ROC}}_n(\alpha)$ approximates $\hat{\epsilon}_n(\alpha)$'s law. The approximation procedure is asymptotically valid, as proved in D.5, so that it satisfies our objective. In practice, one would use $B$ bootstrap samples, get $B$ realizations of $\widehat{\mathrm{ROC}}_n^*(\alpha)$, and thus $B$ realizations of $\hat{\epsilon}_n^{(1)}(\alpha)$ which allow to compute the variability of $\hat{\epsilon}_n^{(1)}(\alpha)$, in order to estimate the variability of $\hat{\epsilon}_n(\alpha)$, *i.e.* the uncertainty of the ROC curve. The *bootstrap percentile* method enables to build accurate confidence regions for the quantity $\widehat{\mathrm{ROC}}_n$ of interest. Indeed, for a fixed $\alpha \in (0, 1)$, a confidence interval, at level $1 - \alpha_{CI} \in [0, 1]$, around $\widehat{\mathrm{ROC}}_n(\alpha)$ can be obtained by considering the $\frac{\alpha_{CI}}{2}$-th and the $\frac{1-\alpha_{CI}}{2}$-th quantiles of $B$ realizations of $\widehat{\mathrm{ROC}}_n(\alpha) + \hat{\epsilon}_n^{(1)}(\alpha)$. It is worth noticing that, in spite of its asymptotic validity, the method can be seriously compromised in the non asymptotic regime when the distribution of $\widehat{\mathrm{ROC}}_n^*(\alpha)$ is not centered at $\widehat{\mathrm{ROC}}_n(\alpha)$. This is typically the case in the present situation, as depicted in Fig. 2: the $B = 200$ realizations of $\widehat{\mathrm{ROC}}_n^*(\alpha)$ (light blue) are not at all centered around the empirical ROC curve (dark blue), the confidence band formed by the *naive*

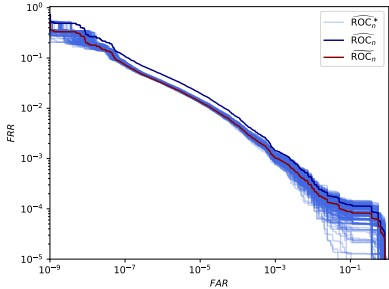
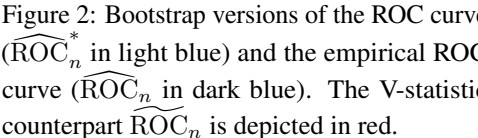
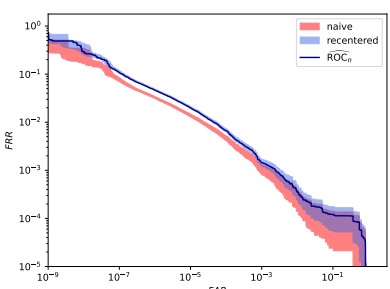

Figure 2: Bootstrap versions of the ROC curve ($\widehat{\text{ROC}}_n^*$ in light blue) and the empirical ROC curve ($\widehat{\text{ROC}}_n$ in dark blue). The V-statistic counterpart $\widetilde{\text{ROC}}_n$ is depicted in red.

Figure 3: Confidence bands at 95% confidence level for the empirical ROC curve (dark blue), using two methods: the naive bootstrap (light red) and the recentered bootstrap (light blue).

*bootstrap*, at level $1 - \alpha_{CI} = 95\%$, being displayed in Fig. 3 (light red). Note that the naive bootstrap of Bertail et al. (2008) yields strongly inaccurate confidence bands (in the non asymptotic regime) as the bands do not even contain the empirical ROC curve most of the time. For both Figures, the pretrained encoder $f$ defining the similarity function $s$ (*cf* Eq. 1) is ArcFace, while MORPH is the evaluation dataset (see 4 for details on models and datasets). We now explain why the distribution of $\widehat{\text{ROC}}_n^*(\alpha)$ is not centered around $\widehat{\text{ROC}}_n(\alpha)$ *i.e.* why the naive bootstrap of Bertail et al. (2008) fails.

**$V$-statistic version of the ROC and recentering.** Firstly, one can easily check that $\mathbb{E}^*[\widehat{\text{FAR}}_n^*(t) \mid \mathcal{D}] = \widehat{\text{FAR}}_n(t)$, where by $\mathbb{E}^*[\cdot \mid \mathcal{D}]$ and $\mathbb{P}^*\{\cdot \mid \mathcal{D}\}$ are meant the conditional expectation and probability given the dataset $\mathcal{D}$ used to compute the empirical criterion $\widehat{\text{ROC}}_n$. Whereas the quantity $\widehat{\text{FAR}}_n^*(t)$ is well centered around the empirical $\widehat{\text{FAR}}_n(t)$ given $\mathcal{D}$, this is not the case for the FRR metric in general. Indeed, we have:

$$\mathbb{E}^* \left[ \widehat{\text{FRR}}_n^*(t) \mid \mathcal{D} \right] = \frac{1}{K} \sum_{k=1}^{K} \frac{1}{n_k^2} \sum_{1 \leq i,j \leq n_k} \mathbb{I}\{s(X_i^{(k)}, X_j^{(k)}) \leq t\} := \widetilde{\text{FRR}}_n(t). \tag{10}$$

The quantity above is an average of $K$ independent $V$-statistics, that may slightly differ from $\widehat{\text{FRR}}_n(t)$, the difference being of order $O(1/n)$. This is due to the presence of the *diagonal terms* $\mathbb{I}\{s(X_i^{(k)}, X_i^{(k)}) \leq t\}$, which results from the fact that perfect similarities (*i.e.* similarities equal to 1 in the cosine similarity (1)) can be observed in the bootstrap samples with non zero probability, while this cannot occur when computing $\widehat{\text{FRR}}_n(t)$. Hence, the empirical FRR tends to be underestimated by its naive bootstrap version in general. This is reflected in the bootstrap ROC curves $\widehat{\text{ROC}}_n^*$ being centered around their V-statistic version $\widetilde{\text{ROC}}_n(\alpha) := \widetilde{\text{FRR}}_n \circ (\widehat{\text{FAR}}_n)^{-1}(\alpha)$, and not around the empirical ROC curve $\widehat{\text{ROC}}_n$, as depicted by Fig. 2. Notice incidentally that this phenomenon is specific to similarity scoring, because of the pairwise nature of the statistic (3), and does not occur in the classic bipartite ranking framework (Bertail et al., 2008). In fact, the V-statistic version of the ROC curve is of great interest since the recentered error $\hat{\epsilon}_n^{(2)}(\alpha) := \widehat{\text{ROC}}_n^*(\alpha) - \widetilde{\text{ROC}}_n(\alpha)$ also approximates $\hat{\epsilon}_n(\alpha)$'s law, as proved in D.5. In the same way than for $\hat{\epsilon}_n^{(1)}(\alpha)$, we are able to build confidence intervals, at confidence level $1 - \alpha_{CI} \in [0, 1]$, for the quantity $\widehat{\text{ROC}}_n(\alpha)$. Considering $B$ bootstrap samples, one would compute $l_{\alpha_{CI}}^{(n,B)}(\alpha)$ (resp. $u_{\alpha_{CI}}^{(n,B)}(\alpha)$) the $\frac{\alpha_{CI}}{2}$-th (resp. the $\frac{1-\alpha_{CI}}{2}$-th) quantile of the $B$ realizations of $\widehat{\text{ROC}}_n(\alpha) + \hat{\epsilon}_n^{(2)}(\alpha)$. $l_{\alpha_{CI}}^{(n,B)}(\alpha)$ and $u_{\alpha_{CI}}^{(n,B)}(\alpha)$ are respectively the lower and upper bounds of the confidence interval, obtained with a variant of the naive bootstrap which we call *recentered bootstrap*. The difference with $\hat{\epsilon}_n^{(1)}(\alpha)$ is that, using Eq. 10, one finds that $\mathbb{E}^* \left[ \widehat{\text{ROC}}_n(\alpha) + \hat{\epsilon}_n^{(2)}(\alpha) \mid \mathcal{D} \right] = \widehat{\text{ROC}}_n(\alpha)$, *i.e.* the confidence intervals are well centered around the empirical ROC curve, as depicted in Fig. 3.

The theoretical considerations within this section can be summarized by the result below. It states that the confidence interval, at confidence level $1 - \alpha_{CI}$, for the ROC curve, using the recentered bootstrap, has a probability of containing the true ROC curve which is truly equal to $1 - \alpha_{CI}$ when $n$ and $B$ tend to $+\infty$. Refer to D.6 for the technical proof. The result holds when applying the recentered bootstrap to the considered fairness metrics (see D.7).

**Theorem 1.** *Let $\alpha \in (0, 1)$ and $\alpha_{CI} \in (0, 1)$. Under the (mild) assumptions in D.4, we have:*

$$\mathbb{P}\{l_{\alpha_{CI}}^{(n,B)}(\alpha) \leq \mathrm{ROC}(\alpha) \leq u_{\alpha_{CI}}^{(n,B)}(\alpha)\} \to 1 - \alpha_{CI},$$

*as $n$ and $B$ both tend to $+\infty$.*

This is an asymptotic result. In the non asymptotic regime, *i.e.* with a finite evaluation dataset, an interesting question is to find what happens to this probability of containing the true ROC curve. For that purpose, it is common practice to use synthetic datasets allowing for an approximation of the true ROC curve. On each dataset, one can compute confidence intervals and check whether the frequency, over all datasets, of containing the true ROC curve is truly equal to $1 - \alpha_{CI}$. In B.1, we generate 200 synthetic datasets and conclude that it is very close to $1 - \alpha_{CI}$, underlining the soundness of the recentered bootstrap, while it is not the case for the naive bootstrap of Bertail et al. (2008).

**Uncertainty metric.** To quantify the uncertainty about the ROC curve, one might be interested in a scalar quantity which summarizes its uncertainty. For instance, $\sqrt{\mathrm{Var}[\hat{\epsilon}_n(\alpha)]}$ seems appropriate for such a quantification. However, in order to compare this scalar uncertainty at several FAR levels $\alpha$, it seems reasonable to consider a *relative* quantity such as $\sqrt{\mathrm{Var}[\hat{\epsilon}_n(\alpha)]}/\widehat{\mathrm{ROC}}_n(\alpha)$. To estimate this quantity, we use the recentered bootstrap and define the *normalized uncertainty* of the ROC curve as:

$$U[\widehat{\mathrm{ROC}}_n(\alpha)] = \frac{\sqrt{\mathrm{Var}[\hat{\epsilon}_n^{(2)}(\alpha) \mid \mathcal{D}]}}{\widehat{\mathrm{ROC}}_n(\alpha)}, \tag{11}$$

as $\hat{\epsilon}_n^{(2)}(\alpha) = \widehat{\mathrm{ROC}}_n^*(\alpha) - \widehat{\mathrm{ROC}}_n(\alpha)$ approximates $\hat{\epsilon}_n(\alpha)$'s law. In practice, one would use $B$ bootstrap samples, get $B$ realizations of $\widehat{\mathrm{ROC}}_n^*(\alpha)$, thus of $\hat{\epsilon}_n^{(2)}(\alpha)$. From those data, one would compute their standard deviation, normalized by the empirical ROC curve. The definition of the normalized uncertainty is naturally extended to fairness metrics (see D.7).

The pseudo-codes for the naive/recentered bootstrap methods, the computation of confidence intervals, as well as of the normalized uncertainty for the ROC curve and fairness metrics are available in C.

## 4 NUMERICAL EXPERIMENTS - APPLICATIONS

**Models and datasets.** We take as encoder $f$ several pre-trained models[1] (AdaCos of Zhang et al. (2019), ArcFace of Deng et al. (2019a), CosFace of Wang et al. (2018), CurricularFace of Huang et al. (2020)) whose backbone is a MobileFaceNet (Chen et al., 2018), trained on the MS-Celeb-1M-v1c-r dataset[2]. This dataset is a cleaned version of the MS-Celeb1M dataset (Guo et al., 2016) and it contains 3.28M images of 73k identities. We choose the MORPH dataset (Ricanek & Tesafaye, 2006) as evaluation dataset. It is composed of 55k face images from 13k distinct identities. This dataset is widely used for fairness evaluation since it is provided with ground-truth age and gender labels (the available labels for the latter are female and male). All images are pre-processed by the Retina-Face detector (Deng et al., 2019c) and are of size $112 \times 112$ pixels. Unless specified, all experiments use $B = 200$ bootstrap samples.

To highlight the significance of the tackled problem in this paper, we show in Fig. 1 the empirical ROC curves of three models (ArcFace, CosFace and AdaCos), computed on two distinct datasets. Those datasets are obtained by splitting MORPH in two parts, with the same number of images, each identity being present in both splits (see A.3). A confidence band at $95\%$ confidence level, computed for ArcFace on one split of data, suggests that ArcFace and CosFace are indistinguishable in terms of performance, for the FAR levels displayed on the $x$-axis. This insight is interesting as there is no model between ArcFace and CosFace that performs better than the other on both datasets.

---

[1] `https://github.com/JDAI-CV/FaceX-Zoo/blob/main/training_mode/README.md`.
[2] See footnote 1.

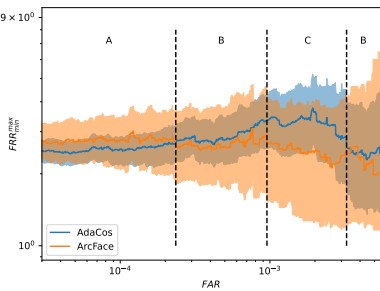
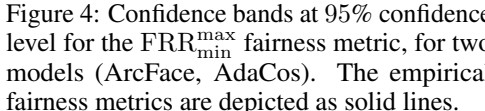

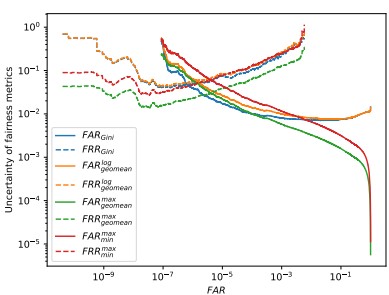

Figure 4: Confidence bands at $95\%$ confidence level for the $\mathrm{FRR}^{\max}_{\min}$ fairness metric, for two models (ArcFace, AdaCos). The empirical fairness metrics are depicted as solid lines.

Figure 5: Normalized uncertainty of several fairness metrics (FAR fairness in solid lines, FRR fairness in dashed lines). The gender label is chosen as the sensitive attribute.

Then, we investigate the uncertainty related to the fairness metric $\mathrm{FRR}^{\max}_{\min}$. The gender label is used here as the sensitive attribute. We display in Figure 4 the confidence bands at $95\%$ confidence level for the $\mathrm{FRR}^{\max}_{\min}$ fairness metric (see B.3 for other fairness metrics), for two models (AdaCos and ArcFace). Three zones (A, B, C) are delimited by dashed lines. For the zone A (resp. C), the empirical fairness is better for AdaCos (resp. ArcFace), while the upper-bound of the confidence band is lower for AdaCos (resp. ArcFace). One would conclude that, for each zone, one model is better than the other in terms of FRR fairness (AdaCos for zone A, ArcFace for zone C). The case of zone B is more complex. Only using the empirical fairness metrics, one would choose ArcFace as the fair model. However, the uncertainty for ArcFace is high, and one may choose AdaCos for its robustness, especially in the case where there would be a strict fairness constraint to deploy the technology (*e.g.* a legislation requiring $\mathrm{FRR}^{\max}_{\min} \leq 4$ at $\mathrm{FAR} = 6 \times 10^{-4}$ for any evaluation dataset).

Finally, we compute the normalized uncertainty of Eq. 11 for all fairness metrics. As illustrated in Figure 5, the max-geomean metric displays (almost always) the lowest uncertainty, both in terms of FAR and FRR, which makes it particularly suitable for fairness evaluation. This finding is supported by similar experiments in B.2, where the trained model, the evaluation dataset and the used sensitive attribute change. In particular, we employ a ArcFace model with a ResNet backbone, evaluated on RFW (Wang et al., 2019). In addition to be more robust than other fairness metrics, the max-geomean metric has the significant advantage to be interpretable.

## 5 CONCLUSION

In this paper, we consider the problem of assessing the uncertainty inherent in estimating ROC curves, using evaluation datasets, in the context of similarity scoring. Quantifying this uncertainty is of great interest since the ROC curve is the gold standard to evaluate performance and fairness in various applications, such as Face Recognition. We show the consistency of empirical similarity ROC curves and propose a variant of the bootstrap approach to build confidence bands around performance/fairness metrics, in order to quantify their variability. The procedure is proved to be asymptotically valid and its relevance is illustrated through applications in Face Recognition. Finally, some popular Face Recognition fairness metrics are compared in terms of their uncertainty, revealing that the max-geomean metric is the more robust to assess fairness. While the gold standard by which fairness will be evaluated in the future is not fixed yet, we believe that it should definitely incorporate uncertainty measures, since it could lead to wrong conclusions otherwise. The bootstrap approach is simple, fast and could greatly improve the reliability of accuracy and fairness metrics, especially within the Face Recognition community.

**Reproducibility.** Pseudo-codes for the experiments of the paper are available in the Supplementary Material C. The open-source pre-trained models used for the experiments are available with download links.

ACKNOWLEDGMENTS

This research was partially supported by the French National Research Agency (ANR), under grant ANR-20-CE23-0028 (LIMPID project).

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
