# A  FURTHER REMARKS

## A.1  FAIRNESS METRICS

In order to inspect the fairness properties of a FR system based on a similarity scoring function $s$, one generally looks at differentials in performance amongst several subgroups/segments of the population. Such subgroups are distinguishable by a *sensitive attribute* (*e.g.* gender, race, age class, ...). For a given (discrete) sensitive attribute that can take $M > 1$ different values, in $\mathcal{A} = \{0, 1, \dots, M - 1\}$ say, we enrich the probability space and now consider a random vector $(X, Y, A)$ where $A \in \mathcal{A}$ indicates the subgroup to which the individual indexed by $Y$ belongs to. For every fixed value $a \in \mathcal{A}$, we can further define the FAR/FRR related to subgroup $a$: $\mathrm{FAR}_a(t) = \mathbb{P}\{s(X, X') > t \mid Y \neq Y', A = A' = a\}$ and $\mathrm{FRR}_a(t) = \mathbb{P}\{s(X, X') \leq t \mid Y = Y', A = A' = a\}$ for all $t \in \mathbb{R}$, where by $(X', Y', A')$ is meant an independent copy of the random triplet $(X, Y, A)$. Ideally, a fair model/function $s$ would exhibit nearly constant $\mathrm{FAR}_a(t)$ values when $a$ varies, for all $t$ (and the same property for the $\mathrm{FRR}_a(t)$ values). A FR fairness metric quantifies how much a model $s$ is far from this property.

In the following, we list several popular FR fairness metrics. All of them are used by the U.S. National Institute of Standards and Technology (NIST) in their FRVT report (Grother, 2022). Those fairness metrics attempt to quantify the differentials in $(\mathrm{FAR}_a(t))_{a \in \mathcal{A}}$ and $(\mathrm{FRR}_a(t))_{a \in \mathcal{A}}$. Each fairness metric has two versions : one for the differentials in terms of FAR, the other for the differentials in terms of FRR.

**Max-min ratio.**  This metric has also been introduced by Conti et al. (2022). Its advantage is to be very interpretable but it is sensitive to low values in the denominator.

$$\mathrm{FAR}_{\min}^{\max}(t) = \frac{\max_{a \in \mathcal{A}} \mathrm{FAR}_a(t)}{\min_{a \in \mathcal{A}} \mathrm{FAR}_a(t)},$$

$$\mathrm{FRR}_{\min}^{\max}(t) = \frac{\max_{a \in \mathcal{A}} \mathrm{FRR}_a(t)}{\min_{a \in \mathcal{A}} \mathrm{FRR}_a(t)}.$$

**Max-geomean ratio.**  This metric replaces the previous minimum by the geometric mean $\mathrm{FAR}^{\dagger}(t)$ of the values $(\mathrm{FAR}_a(t))_{a \in \mathcal{A}}$, in order to be less sensitive to low values in the denominator.

$$\mathrm{FAR}_{\mathrm{geomean}}^{\max}(t) = \frac{\max_{a \in \mathcal{A}} \mathrm{FAR}_a(t)}{\mathrm{FAR}^{\dagger}(t)},$$

$$\mathrm{FRR}_{\mathrm{geomean}}^{\max}(t) = \frac{\max_{a \in \mathcal{A}} \mathrm{FRR}_a(t)}{\mathrm{FRR}^{\dagger}(t)}.$$

**Log-geomean sum.**  It is a sum of normalized logarithms.

$$\mathrm{FAR}_{\mathrm{geomean}}^{\log}(t) = \sum_{a \in \mathcal{A}} \left| \log_{10} \frac{\mathrm{FAR}_a(t)}{\mathrm{FAR}^{\dagger}(t)} \right|,$$

$$\mathrm{FRR}_{\mathrm{geomean}}^{\log}(t) = \sum_{a \in \mathcal{A}} \left| \log_{10} \frac{\mathrm{FRR}_a(t)}{\mathrm{FRR}^{\dagger}(t)} \right|.$$

**Gini coefficient.**  The Gini coefficient is a measure of inequality in a population. It ranges from a minimum value of zero, when all individuals are equal, to a theoretical maximum of one in an infinite population in which every individual except one has a size of zero. In this case, the mean $\mathrm{FAR}^{\diamond}(t)$ that is used is the arithmetic mean of the values $(\mathrm{FAR}_a(t))_{a \in \mathcal{A}}$.

$$\mathrm{FAR}_{\mathrm{Gini}}(t) = \frac{|\mathcal{A}|}{|\mathcal{A}| - 1} \frac{\sum_{a \in \mathcal{A}} \sum_{b \in \mathcal{A}} |\mathrm{FAR}_a(t) - \mathrm{FAR}_b(t)|}{2 |\mathcal{A}|^2 \mathrm{FAR}^{\diamond}(t)},$$

$$\mathrm{FRR}_{\mathrm{Gini}}(t) = \frac{|\mathcal{A}|}{|\mathcal{A}| - 1} \frac{\sum_{a \in \mathcal{A}} \sum_{b \in \mathcal{A}} |\mathrm{FRR}_a(t) - \mathrm{FRR}_b(t)|}{2 |\mathcal{A}|^2 \mathrm{FRR}^{\diamond}(t)}.$$

Note that it is common to plot those fairness metrics not as functions of the threshold $t$ but as functions of the level of FAR associated to the threshold (as for the ROC curve), meaning that the

threshold $t$ is set so that it achieves a $\mathrm{FAR}(t) = \alpha \in (0, 1)$ (for the global/total population, and not for some specific subgroup). In this sense, one would replace the threshold $t$ in each fairness metric by $\mathrm{FAR}^{-1}(\alpha)$. For instance, the strict definition of the max-min fairness metric is, for any $\alpha \in (0, 1)$:

$$\mathrm{FAR}_{\min}^{\max}(\alpha) = \frac{\max_{a \in \mathcal{A}} \mathrm{FAR}_a \circ \mathrm{FAR}^{-1}(\alpha)}{\min_{a \in \mathcal{A}} \mathrm{FAR}_a \circ \mathrm{FAR}^{-1}(\alpha)},$$

$$\mathrm{FRR}_{\min}^{\max}(\alpha) = \frac{\max_{a \in \mathcal{A}} \mathrm{FRR}_a \circ \mathrm{FAR}^{-1}(\alpha)}{\min_{a \in \mathcal{A}} \mathrm{FRR}_a \circ \mathrm{FAR}^{-1}(\alpha)}.$$

For the sake of clarity, we listed the fairness metrics without the pseudo-inverse $\mathrm{FAR}^{-1}$. However, the theoretical results within this paper take into account those strict definitions.

**Remark 2.** *Conti et al. (2022) argue that the choice of a threshold $t$ achieving a global $\mathrm{FAR}(t) = \alpha$ is not entirely relevant since it depends on the relative proportions of each sensitive attribute value $a$ within the evaluation dataset together with the relative proportion of intra-group negative pairs. They propose instead a threshold $t$ such that each group $a$ satisfies $\mathrm{FAR}_a(t) \le \alpha$, for the max-min fairness metrics $\mathrm{FAR}_{\min}^{\max}(t)$ and $\mathrm{FRR}_{\min}^{\max}(t)$. Since we are dealing with a unique evaluation dataset, we do not use such a threshold choice, to be consistent with the other three fairness metrics (max-geomean, log-geomean and Gini) which employ a threshold $t$ such that $\mathrm{FAR}(t) = \alpha$.*

Other fairness metrics exist in the literature such as the maximum difference in the values $(\mathrm{FAR}_a(t))_{a \in \mathcal{A}}$ used by Alasadi et al. (2019) and Dhar et al. (2021). They have the disadvantage of not being normalized and are thus not interpretable, especially when comparing their values at different levels $\alpha$.

## A.2 U-STATISTICS

In the following, we recall the definition of a generalized $U$-statistic and show that empirical versions $\widehat{\mathrm{FAR}}_n(t), \widehat{\mathrm{FRR}}_n(t)$ of the unknown metrics $\mathrm{FAR}(t), \mathrm{FRR}(t)$ are such $U$-statistics.

**Definition 1.** (GENERALIZED $U$-STATISTIC) *Let $K \ge 1$ and $(d_1, \ldots, d_K) \in \mathbb{N}^{*K}$. Let $(X_1^{(k)}, \ldots, X_{n_k}^{(k)})$, $1 \le k \le K$, be $K$ independent samples of i.i.d. random variables, taking their values in some space $\mathcal{X}_k$ with distribution $F_k(dx)$ respectively. The generalized (or $K$-sample) $U$-statistic of degree $(d_1, \ldots, d_K)$ with kernel $H : \mathcal{X}_1^{d_1} \times \cdots \times \mathcal{X}_K^{d_K} \to \mathbb{R}$, square integrable with respect to the probability distribution $\mu = F_1^{\otimes d_1} \otimes \cdots \otimes F_K^{\otimes d_K}$, is defined as*

$$U_{\mathbf{n}}(H) = \frac{1}{\prod_{k=1}^K \binom{n_k}{d_k}} \sum_{I_1} \cdots \sum_{I_K} H(\mathbf{X}_{I_1}^{(1)}; \mathbf{X}_{I_2}^{(2)}; \ldots; \mathbf{X}_{I_K}^{(K)}), \tag{12}$$

*where the symbol $\sum_{I_k}$ refers to summation over all $n_k!/(d_k!(n_k - d_k)!)$ subsets $\mathbf{X}_{I_k}^{(k)} = (X_{i_1}^{(k)}, \ldots, X_{i_{d_k}}^{(k)})$ related to a set $I_k$ of $d_k$ indexes $1 \le i_1 < \ldots < i_{d_k} \le n_k$. It is said symmetric when $H$ is permutation symmetric in each set of $d_k$ arguments $\mathbf{X}_{I_k}^{(k)}$.*

Recall that $\widehat{\mathrm{FAR}}_n(t), \widehat{\mathrm{FRR}}_n(t)$ are defined, for all $t \in \mathbb{R}$, as:

$$\widehat{\mathrm{FAR}}_n(t) = \frac{2}{K(K-1)} \sum_{k<l} \frac{1}{n_k n_l} \sum_{i=1}^{n_k} \sum_{j=1}^{n_l} \mathbb{I}\{s(X_i^{(k)}, X_j^{(l)}) > t\}, \tag{13}$$

$$\widehat{\mathrm{FRR}}_n(t) = \frac{1}{K} \sum_{k=1}^{K} \frac{2}{n_k(n_k-1)} \sum_{1 \le i < j \le n_k} \mathbb{I}\{s(X_i^{(k)}, X_j^{(k)}) \le t\}. \tag{14}$$

Observe that, for a fixed similarity scoring function $s$, the quantity in Equation 14 can be viewed as an average of $K$ independent (mono-sample) non-degenerate $U$-statistics of degree 2 based on the samples $X_1^{(k)}, \ldots, X_{n_k}^{(k)}, 1 \le k \le K$, with symmetric kernel given by:

$$g_t(x, x') = \mathbb{I}\{s(x, x') \le t\} \quad \text{for all } (x, x') \in \mathcal{X}^2.$$

Considering (2) now, it is a $K$-sample $U$-statistic of degree $(1,\ 1,\ \ldots,\ 1)$ with kernel given by:

$$h_t(x_1,\ \ldots,\ x_K) = \frac{2}{K(K-1))} \sum_{k<l}(1 - g_t(x_k, x_l)) \quad \text{for all } (x_1,\ \ldots,\ x_K) \in \mathcal{X}^K.$$

### A.3 MORPH SPLIT

In 4, we explain that the MORPH dataset is split in two parts (only for Fig.1).

The split was made such that each part has the same number of images ($n = 27.338$) and the same identities. We discarded all identities having only one image in the original MORPH dataset, which led to remove $400$ images. Then, for identities having an even number of images, we randomly chose half of them to end up in one split, and the other half in the other split. For identities having an odd number of images, we do the same but we give an extra image to one dataset: for the next identity having an odd number of images, the extra image is given to the other dataset and we repeat the process.

## B ADDITIONAL EXPERIMENTS

### B.1 COVERAGE OF THE CONFIDENCE BANDS

In order to evaluate the soundness of any method to build confidence bands, it is common practice to compute the estimated coverage of the bands. Specifying the confidence level $1 - \alpha_{CI}$ (also called the *nominal coverage*) for our bands should lead to bands having truly a probability equal to $1 - \alpha_{CI}$ to contain the true quantity $\mathrm{ROC}(\alpha)$. To confirm whether this is the case, one may use $N_d$ synthetic datasets. On each dataset separately, the confidence bands are computed and the estimated coverage is the proportion of those $N_d$ confidence bands which contains the true ROC curve. Note that this true ROC curve is impossible to know with real data but may be computed/approximated with synthetic data. Ideally, the estimated coverage should be close to the nominal coverage. In this case, the method for building the confidence intervals is said to *achieve nominal coverage*.

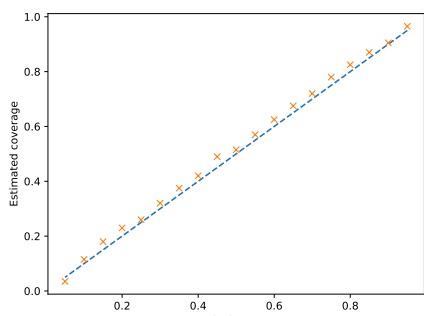

Figure 6: Estimated coverage of our confidence band method for the ROC curve evaluated at $\mathrm{FAR} = 10^{-5}$ (orange). The blue dashed line represents the theoretical target.

**Synthetic data.** Since our method to build confidence bands directly acts on embeddings $f(x_i) \in \mathbb{R}^p$ for a trained encoder $f$, defining the similarity $s$ (Equation 1), the simplest way to create synthetic datasets is to generate synthetic embeddings. It turns out that a natural statistical model for FR embeddings is the mixture of von Mises-Fisher (vMF) distributions, where each component of the mixture is associated to one identity of the dataset (see *e.g.* Hasnat et al. (2017) or Conti et al. (2022)). In details, the embeddings sharing a same identity are modelled as realizations of one vMF distribution, a gaussian in dimension $p$ projected onto the unit hypersphere of dimension $p$, characterized by its centroid (its mean vector) and its concentration parameter (the inverse of the variance). We have randomly drawn $K = 10^3$ centroids on a hypersphere of dimension $p = 128$ using the Marsaglia method (Marsaglia, 1972). The concentration parameter of each vMF distribution is drawn uniformly in $[100, 800]$. Those centroids and concentration parameters fully define our $K$ synthetic identities. A synthetic dataset is generated by sampling $n_k = 10$ points from the vMF distribution (Kim, 2021) associated to identity $k$, for each identity $k$ within the $K$ identities. We obtain a synthetic dataset of $n = \sum_{k=1}^{K} n_k = 10^4$ embeddings. The process is repeated to generate $N_d = 200$ synthetic datasets, sharing the same $K$ identities (*i.e.* the centroids and concentration parameters are shared by all datasets).

**Protocol.** For each dataset, we use our recentered bootstrap method (with $B = 200$ bootstrap samples) to compute confidence intervals, at a given nominal coverage $1 - \alpha_{CI}$, for the ROC curve

evaluated at a FAR level equal to $\alpha = 10^{-5}$. This process results in $N_d$ confidence intervals. To approximate the ground-truth ROC value which those confidence intervals should contain, we concatenate all $N_d$ datasets into one dataset of $N_d \times n = 2 \times 10^6$ embeddings and find its empirical ROC curve $\widehat{\mathrm{ROC}}_n(\alpha)$, still evaluated at a FAR level equal to $\alpha = 10^{-5}$. The estimated coverage is the proportion of the $N_d$ confidence intervals which contains this ground-truth ROC. The result is displayed in Figure 6, for several values of nominal coverage. We list the values displayed on Fig.6 in Table 1. The recentered bootstrap nearly achieves perfect nominal coverage, which supports the soundness of the confidence bands presented within the paper. In other words, the width of each confidence interval is nearly equal to the theoretical target. Theorem 1 claims that this fact is true when the number of evaluation data goes to infinity. In the present case, we demonstrate that the result holds even in the non asymptotic regime.

Table 1: Estimated coverage of the ROC curve evaluated at FAR $= 10^{-5}$, using the recentered bootstrap, for several nominal coverage values.

| Nominal coverage | Estimated coverage |
|---|---|
| 0.95 | 0.96 |
| 0.90 | 0.90 |
| 0.85 | 0.87 |
| 0.80 | 0.82 |
| 0.75 | 0.78 |
| 0.70 | 0.72 |
| 0.65 | 0.67 |
| 0.60 | 0.62 |
| 0.55 | 0.57 |
| 0.50 | 0.51 |
| 0.45 | 0.49 |
| 0.40 | 0.42 |
| 0.35 | 0.37 |
| 0.30 | 0.32 |
| 0.25 | 0.26 |
| 0.20 | 0.23 |
| 0.15 | 0.18 |
| 0.10 | 0.11 |
| 0.05 | 0.04 |

**Naive bootstrap.** We now compute the estimated coverage of the ROC curve, when using the naive bootstrap (Bertail et al., 2008), as a baseline for our method. Computing confidence bands, at nominal coverage $1 - \alpha_{CI} = 95\%$, for the ROC curve on one of the $N_d$ datasets leads to Fig. 7 (using the naive bootstrap) and Fig. 8 (using the recentered bootstrap). As seen with real data in Fig.3, the naive bootstrap underestimates the ROC curve, and thus the confidence bands. In Fig.7-8, the ground-truth ROC curve (not the empirical one) is depicted in dashed lines. Note that the confidence band from the naive bootstrap does not contain the ground-truth ROC for many FAR values. This leads to an estimated coverage for the naive bootstrap that is experimentally equal to zero, no matter what is the confidence level/nominal coverage. For a real comparison between the naive and recentered bootstraps, we choose a FAR value for which the naive bootstrap has chances of being accurate. In this sense, we do not consider the ROC evaluated at FAR $= 10^{-5}$, as in Fig.6, but rather at FAR $= 10^{-1}$, where the ground-truth ROC seems to be contained within the bands of the naive bootstrap (see Fig. 7). We employ the same protocol to compute the estimated coverage as for Fig. 6. The results are presented in Fig. 9 for the naive and the recentered bootstraps in particular, with nominal coverage values leading to a non-zero estimated coverage for the naive bootstrap. We list the values displayed on Fig.9 in Table 2. Note that the naive bootstrap (Bertail et al., 2008) is far from achieving nominal coverage. This underlines the necessity of the recentering step in the bootstrap methodology.

**Gaussian approximation.** In order to provide a more meaningful baseline than the naive bootstrap (Bertail et al., 2008), we design in the following another confidence band method. This method is significantly close to the recentered bootstrap, as there is no such method in the literature for

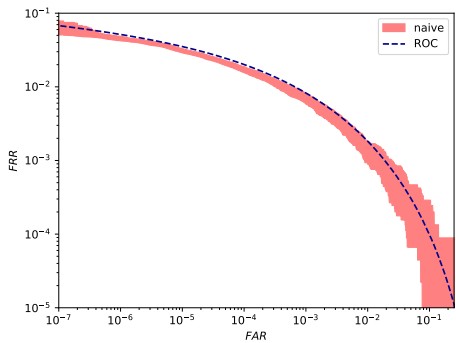 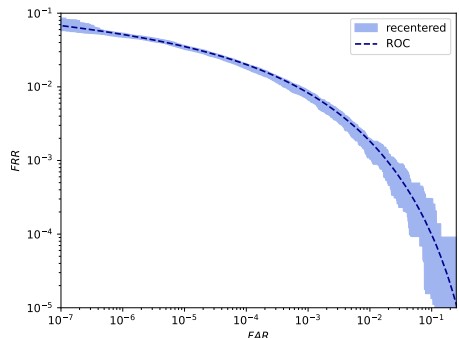

Figure 7: Confidence bands obtained by the naive bootstrap for the ROC curve on one of the $N_d$ synthetic datasets. The ground-truth ROC curve is depicted in dashed lines.

Figure 8: Confidence bands obtained by the recentered bootstrap for the ROC curve on one of the $N_d$ synthetic datasets. The ground-truth ROC curve is depicted in dashed lines.

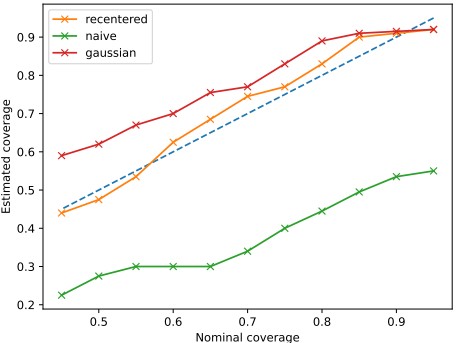

Figure 9: Estimated coverage of three confidence band methods for the ROC curve evaluated at FAR $= 10^{-1}$. The blue dashed line represents the theoretical target. Three methods to build a confidence interval are depicted: the recentered bootstrap (orange), the naive bootstrap (green) and a gaussian approximation of the recentered bootstrap (red).

Table 2: Estimated coverage of the ROC curve evaluated at FAR $= 10^{-1}$, using three confidence band methods, for several nominal coverage values. For each nominal coverage, the best method is displayed in bold, the second best one is underlined.

| Nominal coverage | Estimated coverage | | |
| --- | --- | --- | --- |
| | Recentered bootstrap | Naive bootstrap | Gaussian approximation |
| 0.95 | **0.92** | 0.55 | **0.92** |
| 0.90 | **0.91** | 0.54 | 0.92 |
| 0.85 | **0.90** | 0.50 | 0.91 |
| 0.80 | **0.83** | 0.45 | 0.89 |
| 0.75 | **0.77** | 0.40 | 0.83 |
| 0.70 | **0.74** | 0.34 | 0.77 |
| 0.65 | **0.68** | 0.30 | 0.76 |
| 0.60 | **0.62** | 0.30 | 0.70 |
| 0.55 | **0.53** | 0.30 | 0.67 |
| 0.50 | **0.48** | 0.28 | 0.62 |
| 0.45 | **0.44** | 0.23 | 0.59 |

similarity scoring tasks. The process is almost the same than for Algorithm 2 (recentered bootstrap) where one forms bootstrap samples, then computes a bootstrap version $\widehat{\mathrm{ROC}}_n^*$. Using the recentering $\widetilde{\mathrm{ROC}}_n$, one gets recentered bootstrap versions of the ROC curve, defined as $\widetilde{\mathrm{ROC}}_n(\alpha) + \hat{\epsilon}_n^{(2)}(\alpha)$ where $\hat{\epsilon}_n^{(2)}(\alpha) = \widehat{\mathrm{ROC}}_n^*(\alpha) - \widetilde{\mathrm{ROC}}_n(\alpha)$ (see 3.2). Algorithm 2 then defines the confidence bands as the $\frac{\alpha_{CI}}{2}$-th and $\frac{1-\alpha_{CI}}{2}$-th quantiles of those recentered bootstrap versions of the ROC. Instead of considering those quantiles, a simple idea would be to model $\widetilde{\mathrm{ROC}}_n(\alpha) + \hat{\epsilon}_n^{(2)}(\alpha)$ as a gaussian random variable. As for Algorithm 2 where one simulates $B$ realizations of $\hat{\epsilon}_n^{(2)}(\alpha)$, the same process is applied here to estimate the parameters (mean and variance) of the gaussian distribution of $\widetilde{\mathrm{ROC}}_n(\alpha) + \hat{\epsilon}_n^{(2)}(\alpha)$. Once the parameters estimated, this gaussian approximation allows us to define the bounds of the confidence interval as the $\frac{\alpha_{CI}}{2}$-th and $\frac{1-\alpha_{CI}}{2}$-th quantiles of the corresponding gaussian law. Note that this approximation relies a lot on the recentering which we present within the paper. The estimated coverage of the ROC curve using this gaussian approximation is displayed in Fig. 9 and Table 2. This baseline outperforms significantly the naive bootstrap (Bertail et al., 2008), while being slightly less accurate than our recentered bootstrap. A disadvantage of this method, compared to the recentered bootstrap, is that the confidence intervals are too wide, as seen in Fig. 9 (the estimated coverage in red is most of the time higher than the nominal coverage). Having too wide confidence intervals has its flaws, as one could claim that two models are indistinguishable in terms of performance (ROC) whereas they are not. Confidence bands must be precise, proven valid, in order to be meaningful.

## B.2 COMPARISON OF THE UNCERTAINTY OF FAIRNESS METRICS

In this section, we compute the normalized uncertainty (Eq. 11) for the considered FR fairness metrics, as for Fig.5. In Fig. 10 and 11, we display those uncertainty metrics computed on the MORPH dataset, with the gender label as the sensitive attribute, for two distinct models (CurricularFace and CosFace).

In Fig. 12 and 13, we display those uncertainty metrics computed on the MORPH dataset, with the age label as the sensitive attribute, for two distinct models (CurricularFace and CosFace). The discrete age labels provided by the MORPH dataset are age group labels: '<20', '20-29', '30-39', '40-49', '50+'.

Lastly, we change the evaluation dataset and the backbone of the pre-trained model. We use as encoder the trained[3] model ArcFace (Deng et al., 2019a) whose CNN architecture is a ResNet100 (Han et al., 2017). It has been trained on the MS1M-RetinaFace dataset, introduced by (Deng et al., 2019b) in the ICCV 2019 Lightweight Face Recognition Challenge. We choose the dataset RFW (Wang et al., 2019) as evaluation dataset. It is composed of 40k face images from 11k distinct identities. This dataset is also provided with ground-truth race labels (the four available labels are: African, Asian, Caucasian, Indian) and is widely used for fairness evaluation since it is equally distributed among the race subgroups, in terms of images and identities. The official RFW protocol only considers a few matching pairs among all the possible pairs given the whole RFW dataset. The number of images is typically not enough to get good estimates of our fairness metrics at low FAR. To overcome this, we consider all possible same-race matching pairs among the whole RFW dataset. In Fig.14, we use ArcFace with ResNet100 backbone evaluated on RFW with the sensitive attribute being the race label.

## B.3 OTHER FAIRNESS METRICS FOR FIGURE 4

In Fig. 4, we provide a use-case of model selection depending on a strict fairness constraint. The uncertainty of the fairness metric $\mathrm{FRR}_{\mathrm{min}}^{\mathrm{max}}$ can be high for some models as it can be low. A strict fairness constraint may forbid some models, given their high fairness uncertainty.

We illustrated this use-case with the fairness metric $\mathrm{FRR}_{\mathrm{min}}^{\mathrm{max}}$. In this section, we provide the equivalent of Fig. 4, but for all FRR fairness metrics, for completeness. Those equivalents are displayed in Fig. 16 ($\mathrm{FRR}_{\mathrm{geomean}}^{\mathrm{max}}$), Fig. 17 ($\mathrm{FRR}_{\mathrm{geomean}}^{\mathrm{log}}$) and Fig. 18 ($\mathrm{FRR}_{\mathrm{Gini}}$). As for Fig. 4, the

---

[3] https://github.com/deepinsight/insightface/tree/master/recognition/arcface_torch.

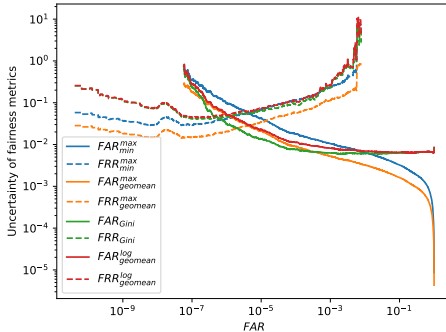

Figure 10: Normalized uncertainty of several fairness metrics (FAR fairness in solid lines, FRR fairness in dashed lines). The dataset is MORPH, the sensitive attribute is the gender label and the encoder $f$ is CurricularFace.

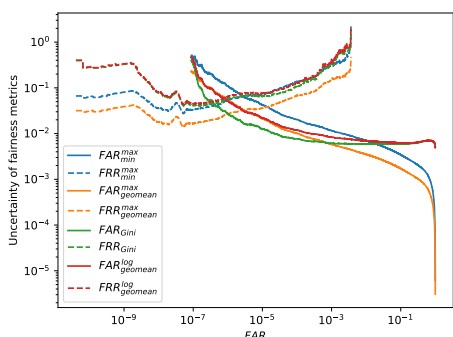

Figure 11: Normalized uncertainty of several fairness metrics (FAR fairness in solid lines, FRR fairness in dashed lines). The dataset is MORPH, the sensitive attribute is the gender label and the encoder $f$ is CosFace.

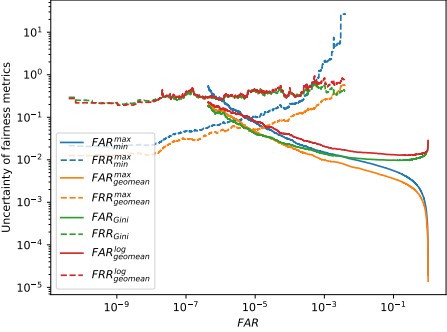

Figure 12: Normalized uncertainty of several fairness metrics (FAR fairness in solid lines, FRR fairness in dashed lines). The dataset is MORPH, the sensitive attribute is the age label and the encoder $f$ is CurricularFace.

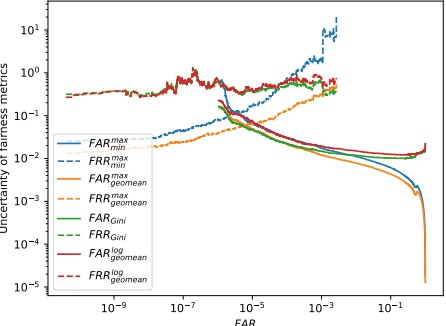

Figure 13: Normalized uncertainty of several fairness metrics (FAR fairness in solid lines, FRR fairness in dashed lines). The dataset is MORPH, the sensitive attribute is the age label and the encoder $f$ is CosFace.

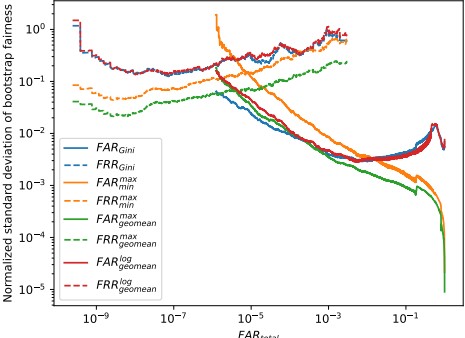

Figure 14: Normalized uncertainty of several fairness metrics (FAR fairness in solid lines, FRR fairness in dashed lines). The dataset is RFW, the sensitive attribute is the race label and the encoder $f$ is ArcFace with ResNet backbone.

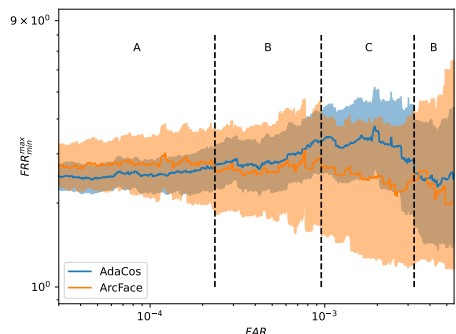
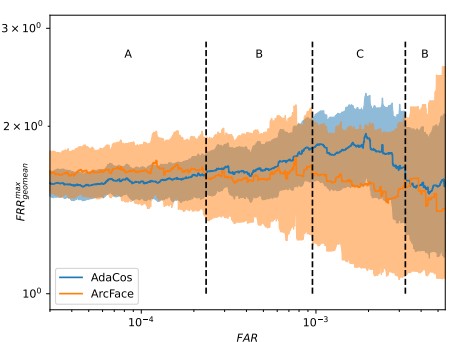

Figure 15: Confidence bands at 95% confidence level for the $\mathrm{FRR}_{\min}^{\max}$ fairness metric, for two models (ArcFace, AdaCos). The empirical fairness metrics are depicted as solid lines.

Figure 16: Confidence bands at 95% confidence level for the $\mathrm{FRR}_{\mathrm{geomean}}^{\max}$ fairness metric, for two models (ArcFace, AdaCos). The empirical fairness metrics are depicted as solid lines.

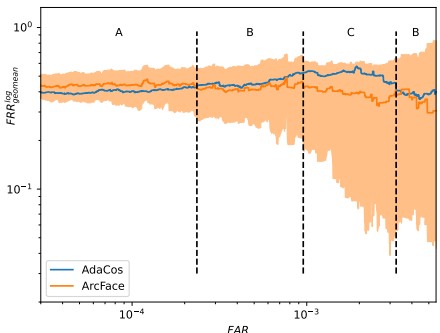
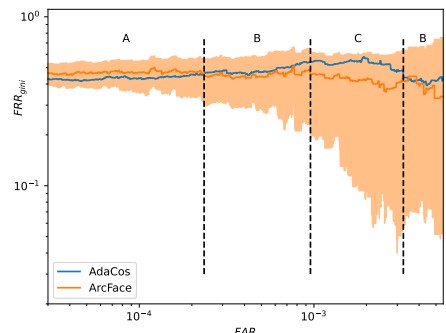

Figure 17: Confidence bands at 95% confidence level for the $\mathrm{FRR}_{\mathrm{geomean}}^{\log}$ fairness metric, for two models (ArcFace, AdaCos). The empirical fairness metrics are depicted as solid lines.

Figure 18: Confidence bands at 95% confidence level for the $\mathrm{FRR}_{\mathrm{Gini}}$ fairness metric, for two models (ArcFace, AdaCos). The empirical fairness metrics are depicted as solid lines.

evaluation dataset is MORPH, the sensitive attribute is the gender label and the number of bootstrap samples given to Algorithm 2 is $B = 200$.

Note that the conclusions from Fig. 4 are unchanged when considering other fairness metrics. Indeed, the fairness metrics of AdaCos and ArcFace all exhibit exactly the same behaviour : both empirical fairness metrics intersect at the same FAR level for all fairness metrics, and the upper bounds of both fairness metrics also intersect at the same FAR levels for all fairness metrics. This striking fact highlights the fact that all fairness metrics measure the same performance differentials. However, one significant difference between those metrics is their uncertainty: the width of the confidence bands can be high, relatively to the value of the empirical fairness, making the fairness metric not so robust. The comparison of the uncertainty between fairness metrics is displayed in Fig. 5 and in Section B.2.

### B.4 FAIRNESS METRICS ON RFW DATASET

For comprehensiveness, we give the confidence bands for all eight considered fairness metrics, listed in section A.1 (see Figures 19, 20, 21, 22). This is the result of Algorithm 2 applied to each fairness metric. We chose the RFW dataset (Wang et al., 2019), as it is balanced in race labels. We already

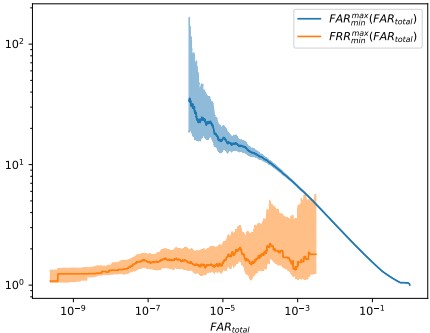 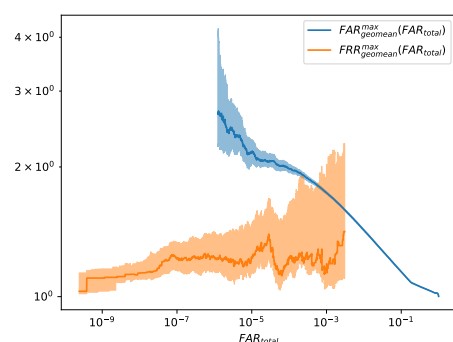

Figure 19: Confidence bands at 95% confidence level for the $\mathrm{FAR}_{\min}^{\max}$ and $\mathrm{FRR}_{\min}^{\max}$ fairness metrics. $B = 100$ bootstrap samples are used. The empirical fairness metrics are depicted as solid lines.

Figure 20: Confidence bands at 95% confidence level for the $\mathrm{FAR}_{\mathrm{geomean}}^{\max}$ and $\mathrm{FRR}_{\mathrm{geomean}}^{\max}$ fairness metrics. $B = 100$ bootstrap samples are used. The empirical fairness metrics are depicted as solid lines.

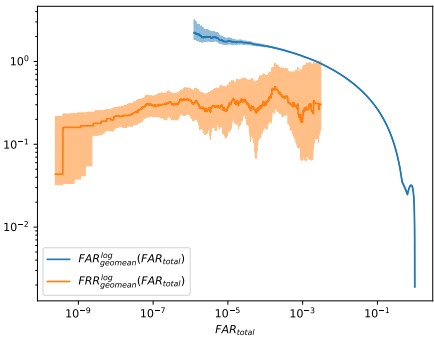 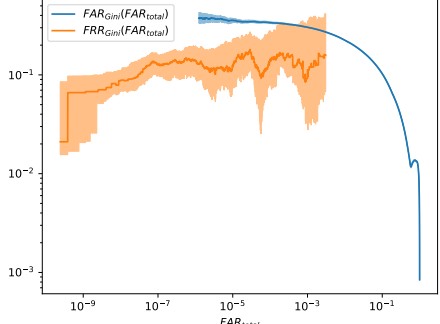

Figure 21: Confidence bands at 95% confidence level for the $\mathrm{FAR}_{\mathrm{geomean}}^{\log}$ and $\mathrm{FRR}_{\mathrm{geomean}}^{\log}$ fairness metrics. $B = 100$ bootstrap samples are used. The empirical fairness metrics are depicted as solid lines.

Figure 22: Confidence bands at 95% confidence level for the $\mathrm{FAR}_{\mathrm{Gini}}$ and $\mathrm{FRR}_{\mathrm{Gini}}$ fairness metrics. $B = 100$ bootstrap samples are used. The empirical fairness metrics are depicted as solid lines.

used the gender and age labels within the paper. The model is ArcFace and we evaluate its fairness at each $\mathrm{FAR}_{\mathrm{total}} = \alpha \in [0, 1]$ level.

## C  PSEUDO-CODE

### C.1  PSEUDO-CODE FOR THE NAIVE BOOTSTRAP

Algorithm 1 provides a pseudo-code for the naive bootstrap of $\widehat{\mathrm{ROC}}_n(\alpha)$, described in section 3.2. The idea is to sample with replacement $n_k$ images among the images of identity $k$ (the i.i.d. variables $X_1^{(k)*}, \ldots, X_{n_k}^{(k)*}$ with distribution $\hat{F}_k$) for each identity $k = 1, \ldots, K$. With those new data, one is able to compute $\widehat{\mathrm{FAR}}_n^*(t)$ and $\widehat{\mathrm{FRR}}_n^*(t)$ from Eq. 8 and 9, and thus a bootstrap version $\widehat{\mathrm{ROC}}_n^*(\alpha)$ of the ROC curve.

By repeating this process $B$ times, one is able to better estimate the variability of $\widehat{\mathrm{ROC}}_n(\alpha)$. At each step $b$, for $b = 1, \ldots, B$, one creates a bootstrap sample $X_{(b)}$ of image data and ends up with a bootstrap version $\widehat{\mathrm{ROC}}_{n,(b)}^*(\alpha)$ of the empirical $\widehat{\mathrm{ROC}}_n(\alpha)$. Accumulating those values for many steps $b$ and for all $\alpha \in (0, 1)$ gives the bundle of light-blue ROC curves in Fig. 2. As explained

in section 3.2, those curves are not centered around the empirical $\widehat{\mathrm{ROC}}_n$, but around its V-statistic counterpart given by $\widetilde{\mathrm{ROC}}_n(\alpha) = \widetilde{\mathrm{FRR}}_n \circ (\widetilde{\mathrm{FAR}}_n)^{-1}(\alpha)$.

---

**Algorithm 1** Naive bootstrap of $\widehat{\mathrm{ROC}}_n(\alpha)$

---

**Input:** $K \geq 1$, images $(X_1^{(1)}, \ldots, X_{n_K}^{(K)})$, encoder $f$
**Require:** $\alpha \in (0, 1)$, $B \geq 1$
**Output:** $B$ naive bootstrap versions $\widehat{\mathrm{ROC}}_n^*(\alpha) = \left(\widehat{\mathrm{ROC}}_{n,(b)}^*(\alpha)\right)_{1 \leq b \leq B}$ of the empirical ROC

 $\widehat{\mathrm{ROC}}_n^*(\alpha) \leftarrow \emptyset$
 **for** $b \leftarrow 1, \ldots, B$ **do**
  $X_{(b)} \leftarrow \emptyset$
  **for** $k \leftarrow 1, \ldots, K$ **do**
   $X_{(b)}^{(k)} \leftarrow$ sample with replacement $n_k$ images among $(X_1^{(k)}, \ldots, X_{n_k}^{(k)})$
   $X_{(b)} \leftarrow X_{(b)} \cup X_{(b)}^{(k)}$
  **end for**
  $\widehat{\mathrm{ROC}}_{n,(b)}^*(\alpha) \leftarrow \widehat{\mathrm{FRR}}_n^* \circ (\widehat{\mathrm{FAR}}_n^*)^{-1}(\alpha)$ for bootstrap sample $X_{(b)}$
  $\widehat{\mathrm{ROC}}_n^*(\alpha) \leftarrow \widehat{\mathrm{ROC}}_n^*(\alpha) \cup \widehat{\mathrm{ROC}}_{n,(b)}^*(\alpha)$
 **end for**
 **return** $\widehat{\mathrm{ROC}}_n^*(\alpha)$

---

**Naive bootstrap of fairness metrics.** Let us take the fairness metric $\mathrm{FRR}_{\min}^{\max}(\alpha)$ as an example. It is defined in section 2.2 as:

$$\mathrm{FRR}_{\min}^{\max}(t) = \frac{\max_{a \in \mathcal{A}} \mathrm{FRR}_a(t)}{\min_{a \in \mathcal{A}} \mathrm{FRR}_a(t)} \quad \text{with } t \text{ such that } \mathrm{FAR}(t) = \alpha.$$

Its empirical version can be defined (see D.3), with the notations $\widehat{\mathrm{FRR}}_{n,a}(t)$ and $\widehat{\mathrm{FAR}}_n(t)$ (defined in Equations 21 and 2), as:

$$\widehat{\mathrm{FRR}}_{\min,n}^{\max}(\alpha) = \frac{\max_{a \in \mathcal{A}} \widehat{\mathrm{FRR}}_{n,a} \circ (\widehat{\mathrm{FAR}}_n)^{-1}(\alpha)}{\min_{a \in \mathcal{A}} \widehat{\mathrm{FRR}}_{n,a} \circ (\widehat{\mathrm{FAR}}_n)^{-1}(\alpha)}.$$

To define the naive bootstrap of the quantity $\mathrm{FRR}_{\min}^{\max}(\alpha)$, it is necessary to first define the naive bootstrap of the quantity $\widehat{\mathrm{FRR}}_{n,a}(t)$ (and $\widehat{\mathrm{FAR}}_{n,a}(t)$ if we consider FAR fairness metrics). The procedure is completely similar to the bootstrap of $\widehat{\mathrm{FRR}}_n(t)$ and $\widehat{\mathrm{FAR}}_n(t)$, presented in section 3.2.

The bootstrap paradigm suggests to recompute the quantities $\widehat{\mathrm{FRR}}_{n,a}(t)$ and $\widehat{\mathrm{FAR}}_{n,a}(t)$ from independent sequences of i.i.d. variables $X_1^{(k)*}, \ldots, X_{n_k}^{(k)*}$ with distribution

$$\hat{F}_k = \frac{1}{n_k} \sum_{i=1}^{n_k} \delta_{X_i^{(k)}}, \tag{15}$$

conditioned upon the original dataset $\mathcal{D} = \{X_1^{(k)}, \ldots, X_{n_k}^{(k)} : k = 1, \ldots, K\}$. In practice of course, the resampling scheme would be applied $B \geq 1$ times in order to compute Monte-Carlo approximations of the distributions of

$$\widehat{\mathrm{FRR}}_{n,a}^*(t) = \frac{1}{K_a} \sum_{\substack{k=1 \\ a_k=a}}^{K} \frac{2}{n_k(n_k-1)} \sum_{1 \leq i < j \leq n_k} \mathbb{I}\{s(X_i^{(k)*}, X_j^{(k)*}) \leq t\}, \tag{16}$$

$$\widehat{\mathrm{FAR}}_{n,a}^*(t) = \frac{2}{K_a(K_a-1)} \sum_{\substack{k<l \\ a_k=a_l=a}} \frac{1}{n_k n_l} \sum_{i=1}^{n_k} \sum_{j=1}^{n_l} \mathbb{I}\{s(X_i^{(k)*}, X_j^{(l)*}) > t\}. \tag{17}$$

Those boostrap versions of $\widehat{\mathrm{FRR}}_{n,a}(t)$ and $\widehat{\mathrm{FAR}}_{n,a}(t)$ allow to define a bootstrap version of each fairness metric. In our running example, the fairness metric $\mathrm{FRR}_{\min}^{\max}(\alpha)$, one may compute its bootstrap version

$$\widehat{\mathrm{FRR}}_{\min,n}^{\max*}(\alpha) = \frac{\max_{a\in\mathcal{A}} \widehat{\mathrm{FRR}}_{n,a}^{*} \circ (\widehat{\mathrm{FAR}}_{n}^{*})^{-1}(\alpha)}{\min_{a\in\mathcal{A}} \widehat{\mathrm{FRR}}_{n,a}^{*} \circ (\widehat{\mathrm{FAR}}_{n}^{*})^{-1}(\alpha)}. \tag{18}$$

The naive bootstrap for the fairness metric $\mathrm{FRR}_{\min}^{\max}(\alpha)$ follows exactly the same algorithm as Algorithm 1, except that the variable $\widehat{\mathrm{ROC}}_{n}^{*}(\alpha)$ is now the bootstrap fairness metric $\widehat{\mathrm{FRR}}_{\min,n}^{\max*}(\alpha)$ and the line $\widehat{\mathrm{ROC}}_{n,(b)}^{*}(\alpha) \leftarrow \widehat{\mathrm{FRR}}_{n}^{*} \circ (\widehat{\mathrm{FAR}}_{n}^{*})^{-1}(\alpha)$ becomes

$$\widehat{\mathrm{FRR}}_{\min,n,(b)}^{\max*}(\alpha) \leftarrow \frac{\max_{a\in\mathcal{A}} \widehat{\mathrm{FRR}}_{n,a}^{*} \circ (\widehat{\mathrm{FAR}}_{n}^{*})^{-1}(\alpha)}{\min_{a\in\mathcal{A}} \widehat{\mathrm{FRR}}_{n,a}^{*} \circ (\widehat{\mathrm{FAR}}_{n}^{*})^{-1}(\alpha)}.$$

### C.2 PSEUDO-CODE FOR THE COMPUTATION OF A CONFIDENCE INTERVAL

Algorithm 2 provides a pseudo-code for the computation of a confidence interval for $\widehat{\mathrm{ROC}}_{n}(\alpha)$ at level $1 - \alpha_{CI}$. It requires the output of Algorithm 1 which gives $B$ naive bootstrap versions $\left(\widehat{\mathrm{ROC}}_{n,(b)}^{*}(\alpha)\right)_{1\leq b\leq B}$ of the empirical ROC. As explained in section 3.2, the recentered bootstrap for $\widehat{\mathrm{ROC}}_{n}(\alpha)$ consists in taking each naive boostrap version $\widehat{\mathrm{ROC}}_{n,(b)}^{*}(\alpha)$, computing its distance to the V-statistic counterpart $\widetilde{\mathrm{ROC}}_{n}(\alpha) = \widetilde{\mathrm{FRR}}_{n} \circ (\widetilde{\mathrm{FAR}}_{n})^{-1}(\alpha)$, and then shifting it by $\widehat{\mathrm{ROC}}_{n}(\alpha)$. This is exactly what Algorithm 2 does, but the shift is done after computation of the quantiles (for the confidence interval). For each naive bootstrap version $\widehat{\mathrm{ROC}}_{n,(b)}^{*}(\alpha)$, we compute the gap with respect to the V-statistic $\widetilde{\mathrm{ROC}}_{n}(\alpha)$, and accumulate, for all bootstrap steps $b$, those distances into the variable gap. Then, a confidence interval at level $1 - \alpha_{CI}$ is computed, giving one lower bound $l$ and one upper bound $u$, defining the confidence interval. Eventually, this confidence interval is shifted by $\widehat{\mathrm{ROC}}_{n}(\alpha)$ as mentioned earlier.

---

**Algorithm 2** Confidence interval for $\widehat{\mathrm{ROC}}_{n}(\alpha)$

---

**Input:** $K \geq 1$, images $(X_1^{(1)}, \ldots, X_{n_K}^{(K)})$, encoder $f$
**Require:** $\alpha \in (0,1)$, $B \geq 1$, $\alpha_{CI} \in (0,1)$
$\quad\quad\quad \widehat{\mathrm{ROC}}_{n}^{*}(\alpha) = \left(\widehat{\mathrm{ROC}}_{n,(b)}^{*}(\alpha)\right)_{1\leq b\leq B}$ from Algorithm 1
**Output:** $l$ and $u$, bounds for the confidence interval of $\widehat{\mathrm{ROC}}_{n}(\alpha)$ at level $1 - \alpha_{CI}$
$\quad \widetilde{\mathrm{ROC}}_{n}(\alpha) \leftarrow \widetilde{\mathrm{FRR}}_{n} \circ (\widetilde{\mathrm{FAR}}_{n})^{-1}(\alpha)$ for original data $(X_1^{(1)}, \ldots, X_{n_K}^{(K)})$
$\quad \mathrm{gap} \leftarrow \emptyset$
$\quad$ **for** $b \leftarrow 1, \ldots, B$ **do**
$\quad\quad \mathrm{gap}_{(b)} \leftarrow \widehat{\mathrm{ROC}}_{n,(b)}^{*}(\alpha) - \widetilde{\mathrm{ROC}}_{n}(\alpha)$
$\quad\quad \mathrm{gap} \leftarrow \mathrm{gap} \cup \mathrm{gap}_{(b)}$
$\quad$ **end for**
$\quad l \leftarrow \frac{\alpha_{CI}}{2}$-th quantile of gap
$\quad u \leftarrow (1 - \frac{\alpha_{CI}}{2})$-th quantile of gap
$\quad \widehat{\mathrm{ROC}}_{n}(\alpha) \leftarrow \widehat{\mathrm{FRR}}_{n} \circ (\widehat{\mathrm{FAR}}_{n})^{-1}(\alpha)$ for original data $(X_1^{(1)}, \ldots, X_{n_K}^{(K)})$
$\quad l \leftarrow \widehat{\mathrm{ROC}}_{n}(\alpha) + l$
$\quad u \leftarrow \widehat{\mathrm{ROC}}_{n}(\alpha) + u$
$\quad$ **return** $l, u$

---

**Confidence interval for fairness metrics.** Let us take the fairness metric $\mathrm{FRR}_{\min}^{\max}(\alpha)$ as an example. It is defined in section 2.2 as:

$$\mathrm{FRR}_{\min}^{\max}(t) = \frac{\max_{a\in\mathcal{A}} \mathrm{FRR}_a(t)}{\min_{a\in\mathcal{A}} \mathrm{FRR}_a(t)} \quad \text{with } t \text{ such that } \mathrm{FAR}(t) = \alpha.$$

Its empirical version can be defined (see D.3), with the notations $\widehat{\mathrm{FRR}}_{n,a}(t)$ and $\widehat{\mathrm{FAR}}_n(t)$ (defined in Equations 21 and 2), as:

$$\widehat{\mathrm{FRR}}_{\mathrm{min,n}}^{\max}(\alpha) = \frac{\max_{a \in \mathcal{A}} \widehat{\mathrm{FRR}}_{n,a} \circ (\widehat{\mathrm{FAR}}_n)^{-1}(\alpha)}{\min_{a \in \mathcal{A}} \widehat{\mathrm{FRR}}_{n,a} \circ (\widehat{\mathrm{FAR}}_n)^{-1}(\alpha)}.$$

To achieve the computation of a confidence interval, the method to do it for the ROC curve, presented above, uses the V-statistic counterpart. In the same way as for $\widehat{\mathrm{FRR}}_n(t)$ (see section 3.2), we can define the V-statistic counterpart for $\widehat{\mathrm{FRR}}_{n,a}(t)$ as

$$\widetilde{\mathrm{FRR}}_{n,a}(t) := \frac{1}{K_a} \sum_{\substack{k=1 \\ a_k=a}}^{K} \frac{1}{n_k^2} \sum_{1 \le i,j \le n_k} \mathbb{I}\{s(X_i^{(k)}, X_j^{(k)}) \le t\}. \tag{19}$$

As for the ROC curve, this allows us to define the V-statistic counterpart of the fairness metric $\mathrm{FRR}_{\min}^{\max}(\alpha)$ as

$$\widetilde{\mathrm{FRR}}_{\mathrm{min,n}}^{\max}(\alpha) = \frac{\max_{a \in \mathcal{A}} \widetilde{\mathrm{FRR}}_{n,a} \circ (\widehat{\mathrm{FAR}}_n)^{-1}(\alpha)}{\min_{a \in \mathcal{A}} \widetilde{\mathrm{FRR}}_{n,a} \circ (\widehat{\mathrm{FAR}}_n)^{-1}(\alpha)}. \tag{20}$$

The computation of a confidence interval for the quantity $\mathrm{FRR}_{\min}^{\max}(\alpha)$ follows exactly the same algorithm as Algorithm 2. We require the output $\widehat{\mathrm{FRR}}_{\mathrm{min},n,(b)}^{\max*}(\alpha)$, instead of $\widehat{\mathrm{ROC}}_{n,(b)}^{*}(\alpha)$, for $b = 1, \ldots, B$, of Algorithm 1 applied to $\mathrm{FRR}_{\min}^{\max}(\alpha)$. For the V-statistic, the line $\widetilde{\mathrm{ROC}}_n(\alpha) \leftarrow \widetilde{\mathrm{FRR}}_n \circ (\widehat{\mathrm{FAR}}_n)^{-1}(\alpha)$ becomes

$$\widetilde{\mathrm{FRR}}_{\mathrm{min,n}}^{\max}(\alpha) \leftarrow \frac{\max_{a \in \mathcal{A}} \widetilde{\mathrm{FRR}}_{n,a} \circ (\widehat{\mathrm{FAR}}_n)^{-1}(\alpha)}{\min_{a \in \mathcal{A}} \widetilde{\mathrm{FRR}}_{n,a} \circ (\widehat{\mathrm{FAR}}_n)^{-1}(\alpha)},$$

hence the gap now measures the distance between $\widehat{\mathrm{FRR}}_{\mathrm{min},n,(b)}^{\max*}(\alpha)$ and $\widetilde{\mathrm{FRR}}_{\mathrm{min,n}}^{\max}(\alpha)$. For the empirical part, the line $\widehat{\mathrm{ROC}}_n(\alpha) \leftarrow \widehat{\mathrm{FRR}}_n \circ (\widehat{\mathrm{FAR}}_n)^{-1}(\alpha)$ becomes

$$\widehat{\mathrm{FRR}}_{\mathrm{min,n}}^{\max}(\alpha) \leftarrow \frac{\max_{a \in \mathcal{A}} \widehat{\mathrm{FRR}}_{n,a} \circ (\widehat{\mathrm{FAR}}_n)^{-1}(\alpha)}{\min_{a \in \mathcal{A}} \widehat{\mathrm{FRR}}_{n,a} \circ (\widehat{\mathrm{FAR}}_n)^{-1}(\alpha)},$$

as well as for the ending lines shifting bounds of the confidence interval.

### C.3 Pseudo-code for the computation of the uncertainty

Algorithm 3 provides a pseudo-code for the computation of the normalized uncertainty (Eq. 11) for the ROC curve. It requires the output of Algorithm 1 which gives $B$ naive bootstrap versions $\left(\widehat{\mathrm{ROC}}_{n,(b)}^{*}(\alpha)\right)_{1 \le b \le B}$ of the empirical ROC. As explained in section 3.2, the standard deviation of each of those $B$ boostrapped values (minus the V-statistic counterpart) is computed and then divided by $\widehat{\mathrm{ROC}}_n(\alpha)$. The algorithm is quite similar to the one used to compute confidence intervals (Algorithm 2).

**Uncertainty of fairness metrics.** The extension to those quantities is completely similar to the extension detailed for Algorithm 2. The computation of the normalized uncertainty for all fairness metrics is provided in Figure 5 and allows to compare the uncertainty of each fairness metric to exhibit their relative robustness.

## D Technical Details

### D.1 A Note on the Pseudo-Inverse of the FAR quantity

In 2.2, we introduced the FAR metric, defined as:

$$\mathrm{FAR}(t) = \mathbb{P}\{s(X, X') > t \mid Z = -1\},$$

---

**Algorithm 3** Uncertainty of $\widehat{\mathrm{ROC}}_n(\alpha)$

---

**Input:** $K \geq 1$, images $(X_1^{(1)}, \ldots, X_{n_K}^{(K)})$, encoder $f$
**Require:** $\alpha \in (0,1)$, $B \geq 1$
    $\widehat{\mathrm{ROC}}_n^*(\alpha) = \big(\widehat{\mathrm{ROC}}_{n,(b)}^*(\alpha)\big)_{1 \leq b \leq B}$ from Algorithm 1
**Output:** $U$, normalized uncertainty of $\widehat{\mathrm{ROC}}_n(\alpha)$
  $\widetilde{\mathrm{ROC}}_n(\alpha) \leftarrow \widehat{\mathrm{FRR}}_n \circ (\widehat{\mathrm{FAR}}_n)^{-1}(\alpha)$ for original data $(X_1^{(1)}, \ldots, X_{n_K}^{(K)})$
  $\mathrm{gap} \leftarrow \emptyset$
  **for** $b \leftarrow 1, \ldots, B$ **do**
    $\mathrm{gap}_{(b)} \leftarrow \widehat{\mathrm{ROC}}_{n,(b)}^*(\alpha) - \widetilde{\mathrm{ROC}}_n(\alpha)$
    $\mathrm{gap} \leftarrow \mathrm{gap} \cup \mathrm{gap}_{(b)}$
  **end for**
  $U \leftarrow$ standard deviation of gap
  $\widehat{\mathrm{ROC}}_n(\alpha) \leftarrow \widehat{\mathrm{FRR}}_n \circ (\widehat{\mathrm{FAR}}_n)^{-1}(\alpha)$ for original data $(X_1^{(1)}, \ldots, X_{n_K}^{(K)})$
  $U \leftarrow U/\widehat{\mathrm{ROC}}_n(\alpha)$
  **return** $U$

---

and the ROC curve as $\mathrm{ROC} \colon \alpha \in (0,1) \mapsto \mathrm{FRR} \circ \mathrm{FAR}^{-1}(\alpha)$.

The pseudo-inverse of any cumulative distribution function (cdf) $\kappa(t)$ on $\mathbb{R}$ is defined as

$$\kappa^{-1}(\alpha) = \inf\{t \in \mathbb{R} : \kappa(t) \geq \alpha\}, \quad \text{for } \alpha \in (0,1).$$

Note that the quantity $\mathrm{FAR}(t)$ is not a cdf, so that its pseudo-inverse is not well defined. However, the opposite of $\mathrm{FAR}(t)$, the True Rejection Rate (TRR), is a proper cdf:

$$\mathrm{TRR}(t) = 1 - \mathrm{FAR}(t) = \mathbb{P}\{s(X, X') \leq t \mid Z = -1\}.$$

As such, the pseudo-inverse $\mathrm{TRR}^{-1}(\alpha)$ is well defined for the TRR quantity. This allows to define the pseudo-inverse for FAR. Indeed, for any $\alpha \in (0,1)$ satisfying $\mathrm{FAR}(t) = \alpha$, one would get the following TRR

$$\mathrm{TRR}(t) = 1 - \alpha.$$

The threshold $t$ of interest is found using the pseudo-inverse of TRR (see Hiesh & Turnbull (1996)):

$$\mathrm{FAR}^{-1}(\alpha) \colon = \mathrm{TRR}^{-1}(1 - \alpha) = (1 - \mathrm{FAR})^{-1}(1 - \alpha).$$

This definition is extended to other quantities within the paper, all being the opposite of one cdf.

### D.2 Proof of Proposition 1 - Consistency of the Empirical Similarity ROC Curve

In the multi-sample asymptotic framework 2.1, by virtue of the $U$-statistic's version of the Strong Law of Large Numbers (see Serfling (1980) or Lee (1990)), we almost-surely have:

$$\widehat{\mathrm{FAR}}_n(t) \to \mathrm{FAR}(t) \text{ and } \widehat{\mathrm{FRR}}_n(t) \to \mathrm{FRR}(t) \text{ as } n \to +\infty,$$

for all $t \in \mathbb{R}$. Applying next the argument of Lemma 21.2 in van der Vaart (1998), we deduce that, for all $\alpha \in (0,1)$, we have with probability one:

$$(\widehat{\mathrm{FAR}}_n)^{-1}(\alpha) \to \mathrm{FAR}^{-1}(\alpha) \text{ as } n \to +\infty.$$

We thus obtain the pointwise consistency of the empirical similarity ROC curve, the uniform version being immediately obtained by a classic Dini's argument, given the monotone nature of (empirical) ROC curves.

Notice incidentally that a bound of order $O_{\mathbb{P}}(1/\sqrt{n})$ could be established by means of the same linearization techniques (*i.e.* Hoeffding decomposition) as those used in Vogel et al. (2018).

## D.3 Definitions of $\widehat{\mathrm{FAR}}_{n,a}(t)$, $\widehat{\mathrm{FRR}}_{n,a}(t)$ and empirical fairness metrics

Without specifying any particular subgroup $a$ (*i.e.* considering the global poulation), we have already explained in section 3.1 that $\widehat{\mathrm{FAR}}_n(t)$ and $\widehat{\mathrm{FRR}}_n(t)$ (defined in Equations 2 and 3) are natural empirical versions of $\mathrm{FAR}(t)$ and $\mathrm{FRR}(t)$. We extend here their definitions to one specific subgroup $a \in \mathcal{A}$. For this purpose, we first define $a_k \in \mathcal{A}$ the (sensitive) attribute label associated with identity $k$, that is the subgroup to which all images of identity $k$ belong to. We also define $K_a$ the number of identities which belong to subgroup $a \in \mathcal{A}$, within the evaluation dataset. For any $a \in \mathcal{A}$, we then define the natural empirical versions of $\mathrm{FAR}_a(t)$ and $\mathrm{FRR}_a(t)$:

$$\widehat{\mathrm{FRR}}_{n,a}(t) = \frac{1}{K_a} \sum_{\substack{k=1 \\ a_k=a}}^{K} \frac{2}{n_k(n_k-1)} \sum_{1 \le i < j \le n_k} \mathbb{I}\{s(X_i^{(k)}, X_j^{(k)}) \le t\}, \quad (21)$$

$$\widehat{\mathrm{FAR}}_{n,a}(t) = \frac{2}{K_a(K_a-1)} \sum_{\substack{k<l \\ a_k=a_l=a}} \frac{1}{n_k n_l} \sum_{i=1}^{n_k} \sum_{j=1}^{n_l} \mathbb{I}\{s(X_i^{(k)}, X_j^{(l)}) > t\}. \quad (22)$$

Note that they are still U-statistics, as for $\widehat{\mathrm{FAR}}_n(t)$ and $\widehat{\mathrm{FRR}}_n(t)$.

The empirical versions of the fairness metrics listed in section A.1 can then be expressed in terms of $\widehat{\mathrm{FAR}}_{n,a}(t)$, $\widehat{\mathrm{FRR}}_{n,a}(t)$ and $\widehat{\mathrm{FAR}}_n(t)$. Let us take the fairness metric $\mathrm{FRR}_{\min}^{\max}(\alpha)$ as an example. It is defined in section A.1 as:

$$\mathrm{FRR}_{\min}^{\max}(\alpha) = \frac{\max_{a\in\mathcal{A}} \mathrm{FRR}_a \circ \mathrm{FAR}^{-1}(\alpha)}{\min_{a\in\mathcal{A}} \mathrm{FRR}_a \circ \mathrm{FAR}^{-1}(\alpha)}.$$

Its empirical version is simply defined as:

$$\widehat{\mathrm{FRR}}_{\min,n}^{\max}(\alpha) = \frac{\max_{a\in\mathcal{A}} \widehat{\mathrm{FRR}}_{n,a} \circ (\widehat{\mathrm{FAR}}_n)^{-1}(\alpha)}{\min_{a\in\mathcal{A}} \widehat{\mathrm{FRR}}_{n,a} \circ (\widehat{\mathrm{FAR}}_n)^{-1}(\alpha)}.$$

## D.4 Assumptions for next results

The following results hold under the classic (mild) assumptions below.

As in D.1, we still define TRR as the cdf associated to the FAR quantity.

**Assumption 1.** *The univariate distributions* TRR *and* FRR *have densities $h$ and $g$ respectively and the slope of the* ROC *curve is bounded:* $\sup_{\alpha\in[0,1]}\{g(\mathrm{TRR}^{-1}(\alpha))/h(\mathrm{TRR}^{-1}(\alpha))\} < \infty$.

**Assumption 2.** *The cdf* TRR *is twice differentiable on $[0,1]$ and $\forall \alpha \in [0,1]$, $h(\alpha) > 0$ and there exists $\gamma > 0$ such that* $\sup_{\alpha\in[0,1]}\{\alpha(1-\alpha) \cdot d\log(h \circ \mathrm{TRR}^{-1}(\alpha))/d\alpha\} \le \gamma < \infty$.

## D.5 Asymptotic validity of the naive and recentered bootstraps

The result below also states the asymptotic validity of the naive bootstrap, described in 3.2.

**Theorem 2.** *Suppose that Assumptions 1-2 are satisfied. Then, for all $\alpha \in (0,1)$, we almost-surely have:*

$$\sup_{v\in\mathbb{R}} \left| \mathbb{P}^* \left\{ \sqrt{n}|\widehat{\mathrm{ROC}}_n^*(\alpha) - \widehat{\mathrm{ROC}}_n(\alpha)| \le v \mid \mathcal{D} \right\} - \mathbb{P} \left\{ \sqrt{n}|\widehat{\mathrm{ROC}}_n(\alpha) - \mathrm{ROC}(\alpha)| \le v \right\} \right| \to 0,$$
$$(23)$$

*as $n \to +\infty$.*

The proof is the same as that of Theorem 3.

The result below states the asymptotic validity of the recentered bootstrap for the ROC curve, described in 3.2. As explained within the proof, the result holds when replacing the ROC curve by fairness metrics. Indeed, one can define the bootstrap fairness metrics, as well as the $V$-statistic version of those metrics, in the same way than for the ROC curve.

**Theorem 3.** *Suppose that Assumptions 1-2 are satisfied. Then, for all $\alpha \in (0,1)$, we almost-surely have:*

$$\sup_{v \in \mathbb{R}} \left| \mathbb{P}^* \left\{ \sqrt{n} |\widehat{\mathrm{ROC}}_n^*(\alpha) - \widetilde{\mathrm{ROC}}_n(\alpha)| \leq v \mid \mathcal{D} \right\} - \mathbb{P} \left\{ \sqrt{n} |\widehat{\mathrm{ROC}}_n(\alpha) - \mathrm{ROC}(\alpha)| \leq v \right\} \right| \to 0,$$
(24)

*as $n \to +\infty$.*

*Proof.* The asymptotic validity of the naive bootstrap procedure for non-degenerate (generalized, multivariate) $U$-statistics is proved in Bickel & Freedman (1981) (see section 3 therein, refer also to Theorem 3 in Janssen (1997) and to Theorem 3.3 in Shao & Tu (1995)). The proof is based on their asymptotic Gaussianity as well as that of the related $V$-statistics in the asymptotic framework 2.1 (refer to *e.g.* Lee (1990)), which holds true under the assumptions stipulated, combined with a coupling argument. Hence, with probability one, the absolute deviation between the bootstrap approximation

$$\mathbb{P}^* \left\{ \sqrt{n} \left( \widehat{\mathrm{FAR}}_n^*(t_1) - \widehat{\mathrm{FAR}}_n(t_1) \right) \leq u, \ \sqrt{n} \left( \widehat{\mathrm{FRR}}_n^*(t_2) - \widetilde{\mathrm{FRR}}_n(t_2) \right) \leq v \mid \mathcal{D} \right\}$$

and the root distribution

$$\mathbb{P} \left\{ \sqrt{n} \left( \widehat{\mathrm{FAR}}_n(t_1) - \mathrm{FAR}(t_1) \right) \leq u, \ \sqrt{n} \left( \widehat{\mathrm{FRR}}_n(t_2) - \mathrm{FRR}(t_2) \right) \leq v \right\}$$

converges to 0, uniformly in $(u,v) \in \mathbb{R}^2$, as $n \to \infty$. As noticed in Janssen (1997), the bootstrapped FRR statistic $\widehat{\mathrm{FRR}}_n^*(t_2)$ can be recentered either by the $V$-statistic version $\widetilde{\mathrm{FRR}}_n(t_2)$ or else by the original $U$-statistic $\widehat{\mathrm{FRR}}_n(t_2)$ due to the non-degeneracy property.

In addition, under the hypotheses stipulated, the asymptotic normality of the bivariate random vectors

$$\sqrt{n} \left( (\widehat{\mathrm{FAR}}_n)^{-1}(\alpha) - \mathrm{FAR}^{-1}(\alpha), \ \widehat{\mathrm{FRR}}_n(t) - \mathrm{FRR}(t) \right)$$

can be classically deduced from that of the random vectors

$$\sqrt{n} \left( \widehat{\mathrm{FAR}}_n(t_1) - \mathrm{FAR}(t_1), \ \widehat{\mathrm{FRR}}_n(t_2) - \mathrm{FRR}(t_2) \right)$$

by means of classic (linearization) arguments for empirical quantiles (see *e.g.* Chapter 21 in van der Vaart (1998)). A Central Limit Theorem for $\sqrt{n}(\widehat{\mathrm{ROC}}_n(\alpha) - \mathrm{ROC}(\alpha))$ can be easily deduced and exactly the same argument as the one developed in Bickel & Freedman (1981) can then be used to prove (24).

In a similar fashion, the asymptotic validity of the naive/recentered bootstrap applied to Hadamard differentiable functionals of the random vector $(\widehat{\mathrm{FAR}}_{n,a}(t), \widehat{\mathrm{FRR}}_{n,a}(t'))_{a \in \mathcal{A}}$, such as the fairness metrics considered here, results from the asymptotic normality property combined with Theorem 23.9 in van der Vaart (1998). $\qquad \square$

### D.6 CONSISTENCY OF THE COVERAGE PROVIDED BY THE CONFIDENCE INTERVALS

It results from Theorem 3.

**Corollary 1.** *Let $\alpha \in (0,1)$ and $\alpha_{CI} \in (0,1)$. Under the assumptions of Theorem 3, we have:*

$$\mathbb{P}\{l_{\alpha_{CI}}^{(n,B)}(\alpha) \leq \mathrm{ROC}(\alpha) \leq u_{\alpha_{CI}}^{(n,B)}(\alpha)\} \to 1 - \alpha_{CI},$$

*as $n$ and $B$ both tend to $+\infty$.*

*Proof.* The consistency of the probability coverages of the Monte Carlo confidence intervals based on $B \geq 1$ bootstrap replicates and described in subsection 3.2 immediately results from Theorem 3, combined with the Strong Law of Large Numbers. $\qquad \square$

### D.7 BOOTSTRAP AND NORMALIZED UNCERTAINTY OF FAIRNESS METRICS

We have illustrated all results with the use case of the ROC curve but those results hold for the fairness metrics listed in A.1. One only has to replace the ROC-related quantities by their fairness counterparts:

- $\text{ROC}(\alpha)$: the true fairness metrics have been defined in A.1.

- $\widehat{\text{ROC}}_n(\alpha)$: the empirical fairness metrics are defined in D.3. Their definition is very similar to $\widehat{\text{ROC}}_n(\alpha)$ in the sense that one has to replace the FAR, FRR quantities by their empirical versions (*plug-in*).

- $\widehat{\text{ROC}}_n^*(\alpha)$: the bootstrap fairness metrics are nothing but the empirical fairness metrics, computed with a bootstrap sample, instead of the original dataset (exactly like the ROC curve).

- $\widetilde{\text{ROC}}_n(\alpha)$: the $V$-statistic version of fairness metrics is simply obtained by replacing $\text{FAR}_a(t)$ with its empirical version $\widehat{\text{FAR}}_{n,a}(t)$ (see D.3) and $\text{FRR}_a(t)$ with its $V$-statistic version, exactly as for the ROC curve.

**Bootstrap and confidence intervals.** As explained within the proof of Theorem 3, the naive/recentered bootstrap validity holds for fairness metrics when replacing the ROC-related quantities with the fairness quantities listed above. The consistence of the confidence intervals (Corollary 1) directly results from Theorem 3.

**Normalized uncertainty.** In 3.2, we defined a scalar uncertainty measure for the ROC curve as:

$$U[\widehat{\text{ROC}}_n(\alpha)] = \frac{\sqrt{\text{Var}[\hat{\epsilon}_n^{(2)}(\alpha) \mid \mathcal{D}]}}{\widehat{\text{ROC}}_n(\alpha)}, \tag{25}$$

with $\hat{\epsilon}_n^{(2)}(\alpha) = \widehat{\text{ROC}}_n^*(\alpha) - \widetilde{\text{ROC}}_n(\alpha)$. Replacing $\widehat{\text{ROC}}_n^*(\alpha)$, $\widetilde{\text{ROC}}_n(\alpha)$ by their fairness counterparts (see above) allows to get a fairness version of $\hat{\epsilon}_n^{(2)}(\alpha)$. Then, to get the normalized uncertainty of a fairness measure, one only has to compute the (root of) variance of this fairness version of $\hat{\epsilon}_n^{(2)}(\alpha)$, and then to normalize by the corresponding empirical fairness measure. As explained within the proof of Theorem 3, the naive/recentered bootstrap validity holds for fairness metrics when replacing the ROC-related quantities with the fairness quantities listed above. Thus, the normalized uncertainty is as pertinent to fairness metrics than it is to the ROC curve.

Note that we give the methodology (bootstrap, confidence intervals, normalized uncertainty) step by step, for one fairness metric, in the pseudo-code C.