# OpenReview forum: "Assessing Uncertainty in Similarity Scoring: Performance & Fairness in Face Recognition"
_ICLR.cc/2024/Conference — ICLR 2024 poster_

### Official Review · Reviewer_XZHc · 2023-10-28

**Soundness:** 3 good
**Presentation:** 3 good
**Contribution:** 3 good
**Rating:** 6
**Confidence:** 4

**Summary:**

This paper proposes a method for evaluating uncertainty on the ROC curve in face recognition problem.

**Strengths:**

The paper provides a theoretical analysis of the properties and uncertainties associated with the ROC curve in face recognition evaluations and introduces a novel method for quantifying this uncertainty.

**Weaknesses:**

1. This paper dedicates excessive length and involves many complex formulas to introduce some basic knowledge of face recognition evaluation, which seems unnecessary.
2. While fairness in face recognition is a pressing issue, the contributions of this paper appear limited. As the authors themselves note, typically, global ROC curves and their comparisons with specific attribute-driven ROC curves would suffice for the uncertainty analysis in most face recognition scenarios.

**Questions:**

None

---

> ### Author Response · Authors · 2023-11-21
> **Answer 1/3**
>
> We would like to thank you for your feedback. Here are our answers to your concerns.
>
> 1. This paper dedicates excessive length and involves many complex formulas to introduce some basic knowledge of face recognition evaluation, which seems unnecessary.
>
> In our opinion, the formulas introduced are necessary since they shed light onto the complexity of the structure of an empirical similarity ROC curve (and that of fairness metrics) in Face Recognition (FR), given an evaluation dataset, which determines the methodology to assess its variability/uncertainty.
>
> For instance, Eq. (2-3) explain the U-statistic nature of the statistics involved in its computation and underlines the differences with the (simpler) related work of Bertail et al. (2008), which deals with basic i.i.d. averages in contrast to the present FR framework (see Section 3.1). In addition, Equations (8), (9) and (10) are at the heart of our method (the recentered bootstrap) and give an insight into the reasons why the recentering technique is necessary, underestimating the FRR metric otherwise (see Section 3.2).
>
> All the notations presented in the paper are involved in our main results (Proposition 1 and Theorem 1) which prove the asymptotic validity of the recentered bootstrap.
>
> Note that we chose to put the majority of formulas within the supplementary material in order to make the reading easier. For instance, the supplementary material contains the list of popular FR fairness metrics, as well as their empirical counterparts, their attribute-driven versions and their bootstrap versions.
>
> 2.1. While fairness in face recognition is a pressing issue, the contributions of this paper appear limited.
>
> First, we present (and give intuitions about) a new bootstrap method, the recentered bootstrap, which can be applied to similarity scoring tasks. Without this recentering, the naive bootstrap of Bertail et al. (2008) significantly fails, as we show within the paper (e.g. Section 3.2). This new bootstrap method allows to compute proven valid confidence intervals for the ROC curve and popular FR fairness metrics (see Theorem 1). This is the first work to tackle this problem, which has significant impact in real-life applications.
>
> Secondly, we highlight that relying on empirical ROC curves (and fairness metrics) solely to compare models is inaccurate: practitioners of Face Recognition (FR) know that the performance measure varies when the evaluation dataset is changed, even if it is drawn in the same population (partially observable unavoidably). Fig. 1 shows that ArcFace is better than CosFace on one dataset and worse than CosFace on the other dataset. Both datasets share the same identities but not the same images. If only one of these datasets is available, one would jump to an erroneous conclusion. The confidence band method we present in the paper avoids such conclusions as both models are indistinguishable in terms of performance. \
> As many recent FR papers [1,2,3,4] improve the state-of-the-art performance (ROC) by very slight margins on one/two specific evaluation datasets, it becomes necessary to question the uncertainty regarding the true performance of these models. One new architecture/loss function might obtain a slightly better empirical ROC curve than other papers, while this empirical ROC might be contained within our ROC confidence bands for those other papers. Incorporating the uncertainty band in future model comparisons would lead to trustworthy conclusions.
>
> Thirdly, we chose to illustrate a model selection based on fairness in Figure 4. In this case, we employ a model deployment/production point of view. Face Recognition already has a considerable societal impact, while being biased against some subgroups of the population. In this sense, FR applications may have strict fairness constraints. For instance, one hypothetical legislation could be that the FRR metric for women should be lower than 4 times the FRR for men. In Figure 4 (zone B, e.g. FAR=6e-4), this constraint would forbid the ArcFace model (the FRR max-min fairness has confidence bands which can go higher than 4), i.e. the fairest model. Indeed, as for Fig. 1, one evaluation dataset might have this fairness metric equal to 4. Between AdaCos and ArcFace, one would then choose AdaCos, which is the model having the worst empirical fairness measure.

---

> ### Author Response · Authors · 2023-11-21
> **Answer 2/3**
>
> Fourthly, a significant contribution we make is the comparison of FR fairness metrics depending on their uncertainty. The question of measuring fairness is complex and the NIST, responsible for fairness audit of academic/industrial FR algorithms, is still hesitant on which fairness metric one should use to compare models [5]. Besides quantifying the differentials in FAR/FRR (among subgroups), such fairness metrics should be robust and exhibit a low uncertainty. Indeed, with high-impact applications, a fairness audit should be as trustworthy as possible and fairness measures should not vary a lot depending on the evaluation dataset. In addition, the subgroups considered become smaller and smaller when considering intersections of attributes (e.g. black women of age 40-50) and evaluation datasets might not have lots of data for such specific subgroups. This results in uncertain measures for those subgroups, altering the robustness of fairness measures. It becomes necessary to design fairness measures which are as robust as possible.
> We compare popular FR fairness measures, used by the NIST, depending on their uncertainty in Fig. 5 and conclude that the max-geomean metric is the more adequate. This result is supported by similar findings in the supplementary material B.2 (when varying the FR models, the sensitive attribute and the evaluation dataset). Note that this uncertainty quantification will be useful when evaluating the soundness of future fairness metrics.
>
> Lastly, we provide in the supplementary material B.1 the coverage of the bands obtained with the recentered bootstrap. This coverage metric is the gold standard to quantify the accuracy of any method which builds confidence bands. The ideal property of a confidence interval is that specifying its confidence level $1-\alpha_{CI}$ (also called the *nominal coverage*) should lead to an interval having truly a probability equal to $1-\alpha_{CI}$ to contain the true (unknown) quantity $\mathrm{ROC}(\alpha)$. The coverage measures what this probability truly is and it should be equal (or very close) to the specified nominal coverage (see B.1). From Fig. 6 and Table 1, we observe that, even with finite datasets, it is the case for the recentered bootstrap, which suggests that the latter provides an appropriate tool for assessing the uncertainty of the performance measure.
>
> We added two baselines for the revised version of the paper, to compare with the rencentered bootstrap.
>
> The first one is the naive bootstrap of Bertail et al. (2008). The non pairwise setup of the latter work significantly underestimates the ROC curve, leading to strongly inaccurate confidence intervals, as seen in Fig. 9 and Table 2.
>
> As pointed out in the related works section of the paper, no method to quantify the uncertainty in measuring performance/fairness in FR is documented in the literature to the best of our knowledge. The one described and analyzed in this paper is the first one. As explained in Section 3.1, using the (Gaussian) asymptotic law of the empirical similarity ROC curve is not an option given the very complex form of its covariance structure (involving probability density functions).
> That being said, we designed a more meaningful baseline than the naive bootstrap of Bertail et al. (2008). The sole reasonable alternative is to use the bootstrap technique we propose to evaluate the variance of the empirical ROC at a certain FAR level and build a Gaussian confidence interval based on it (see supplementary material B.1). As now shown in the revised version (see Fig. 9 and Table 2), the coverage of this Gaussian approximation is much more accurate than the naive bootstrap, but less satisfactory than our recentered bootstrap.
>
> The approach we propose is easy to use and proven valid, theoretically and empirically (very satisfactory coverage). It avoids the bias that would be caused by a naive implementation of the bootstrap technique and has the potential to be widely used by practitioners, insofar as absolutely no alternative is available today.
>
> [1] Huang, Yuge, et al. "Curricularface: Adaptive curriculum learning loss for deep face recognition." Proceedings of the IEEE/CVF conference on computer vision and pattern recognition. 2020.
>
> [2] Meng, Qiang, et al. "Magface: A universal representation for face recognition and quality assessment." Proceedings of the IEEE/CVF conference on computer vision and pattern recognition. 2021.
>
> [3] Boutros, Fadi, et al. "Elasticface: Elastic margin loss for deep face recognition." Proceedings of the IEEE/CVF conference on computer vision and pattern recognition. 2022.
>
> [4] Kim, Minchul, Anil K. Jain, and Xiaoming Liu. "Adaface: Quality adaptive margin for face recognition." Proceedings of the IEEE/CVF conference on computer vision and pattern recognition. 2022.
>
> [5] Grother, Patrick. "Face Recognition Vendor Test (FRVT) Part 8: Summarizing Demographic Differentials." (2022).

---

> ### Author Response · Authors · 2023-11-21
> **Answer 3/3**
>
> 2.2. As the authors themselves note, typically, global ROC curves and their comparisons with specific attribute-driven ROC curves would suffice for the uncertainty analysis in most face recognition scenarios.
>
>
> We think that comparing global ROC curves with specific attribute-driven ROC curves is a poor method of estimating the uncertainty of any algorithm as it depends on the choice/availability of those attribute labels within the evaluation dataset.
>
>
> But more importantly, we have highlighted that relying on empirical ROC curves (and fairness metrics) solely to compare models is inaccurate: practitioners of FR know very well that the performance measure varies when the evaluation dataset is changed, even if it is drawn in the same population (partially observable unavoidably). Fig. 1 shows that ArcFace is better than CosFace on one dataset and worse than CosFace on the other dataset. Both datasets share the same identities but not the same images. If only one of these datasets is available, one would jump to an erroneous conclusion. The confidence band method we present in the paper avoids such conclusions as both models are indistinguishable in terms of performance.
>
> In a similar manner than for Fig. 1, the ranking of the best attribute-driven ROC curves can be permuted when changing the evaluation dataset. Thus, no trustworthy conclusion can be made, when only looking at empirical attribute-driven ROC curves. The only proper way to compare attribute-driven ROC curves is to compute them, along their confidence bands.

---

### Official Review · Reviewer_EFt7 · 2023-10-30

**Soundness:** 3 good
**Presentation:** 2 fair
**Contribution:** 2 fair
**Rating:** 5
**Confidence:** 3

**Summary:**

This paper focuses on the uncertainty of ROC curve in Face Recognition applications based on similarity scoring. It first proves asymptotic consistency guarantees for empirical ROC curves of similarity functions, which is the theoretical basis of the methodology of building confidence bands for ROC. Then, it provides recentering technique to counteract the underestimation of the similarity ROC curve caused by naïve bootstrap. Resulting from this bootstrap variant, the confidence bands and uncertainty measures for the ROC are defined and shown to be consistent. Finally, the results of various experiments using real face image datasets are illustrated to discuss the practical relevance of the proposed methods.

**Strengths:**

1. In some cases, the simple ROC curve is not enough to compare the performance of different FC models, and the idea of introducing uncertainty into the evaluation metric fills this gap and is relatively practical.
2. The proposed method for estimating the confidence intervals and uncertainty of ROC is intuitive and simple, and can be easily applied directly to various Face Recognition models.

**Weaknesses:**

1. This paper is not the first to apply the bootstrap method to ROC curves and there isn’t enough difference between it and Bertail’s work proposed in 2008. The concept of uncertainty proposed in this paper in relatively new in the field of ROC curve estimation, but it is too simplistic in the field if uncertainty estimation, which needs deeper elaboration.
2. The experiments demonstrates the scenarios and ways in which the method proposed in this paper can be applied, but it is not sufficient to illustrate its effectiveness, and there is a lack of experimental evidence that the proposed uncertainty is a better evaluation metric to compare the performance and fairness of Face Recognition models. Besides, there is no comparison of the proposed method with other methodologies.

**Questions:**

1. This paper is not the first to apply the bootstrap method to ROC curves and there isn’t enough difference between it and Bertail’s work proposed in 2008. The concept of uncertainty proposed in this paper in relatively new in the field of ROC curve estimation, but it is too simplistic in the field if uncertainty estimation, which needs deeper elaboration.
2. The experiments demonstrates the scenarios and ways in which the method proposed in this paper can be applied, but it is not sufficient to illustrate its effectiveness, and there is a lack of experimental evidence that the proposed uncertainty is a better evaluation metric to compare the performance and fairness of Face Recognition models. Besides, there is no comparison of the proposed method with other methodologies.

---

> ### Author Response · Authors · 2023-11-21
> **Answer 1/4**
>
> We would like to thank you for your careful reading of the manuscript and your feedback. Here are our answers to your concerns.
>
> 2.1. The experiments demonstrate the scenarios and ways in which the method proposed in this paper can be applied, but it is not sufficient to illustrate its effectiveness
>
> First, we highlight that relying on empirical ROC curves (and fairness metrics) solely to compare models is inaccurate: practitioners of Face Recognition (FR) know that the performance measure varies when the evaluation dataset is changed, even if it is drawn in the same population (partially observable unavoidably). Fig. 1 shows that ArcFace is better than CosFace on one dataset and worse than CosFace on the other dataset. Both datasets share the same identities but not the same images. If only one of these datasets is available, one would jump to an erroneous conclusion. The confidence band method we present in the paper avoids such conclusions as both models are indistinguishable in terms of performance. \
> As many recent FR papers [1,2,3,4] improve the state-of-the-art performance (ROC) by very slight margins on one/two specific evaluation datasets, it becomes necessary to question the uncertainty regarding the true performance of these models. One new architecture/loss function might obtain a slightly better empirical ROC curve than other papers, while this empirical ROC might be contained within our ROC confidence bands for those other papers. Incorporating the uncertainty band in future model comparisons would lead to trustworthy conclusions.
>
> Secondly, we chose to illustrate a model selection based on fairness in Figure 4. In this case, we employ a model deployment/production point of view. Face Recognition already has a considerable societal impact, while being biased against some subgroups of the population. In this sense, FR applications may have strict fairness constraints. For instance, one hypothetical legislation could be that the FRR metric for women should be lower than 4 times the FRR for men. In Figure 4 (zone B, e.g. FAR=6e-4), this constraint would forbid the ArcFace model (the FRR max-min fairness has confidence bands which can go higher than 4), i.e. the fairest model. Indeed, as for Fig. 1, one evaluation dataset might have this fairness metric equal to 4. Between AdaCos and ArcFace, one would then choose AdaCos, which is the model having the worst empirical fairness measure.
>
> Thirdly, a significant contribution we make is the comparison of FR fairness metrics depending on their uncertainty. The question of measuring fairness is complex and the NIST, responsible for fairness audit of academic/industrial FR algorithms, is still hesitant on which fairness metric one should use to compare models [5]. Besides quantifying the differentials in FAR/FRR (among subgroups), such fairness metrics should be robust and exhibit a low uncertainty. Indeed, with high-impact applications, a fairness audit should be as trustworthy as possible and fairness measures should not vary a lot depending on the evaluation dataset. In addition, the subgroups considered become smaller and smaller when considering intersections of attributes (e.g. black women of age 40-50) and evaluation datasets might not have lots of data for such specific subgroups. This results in uncertain measures for those subgroups, altering the robustness of fairness measures. It becomes necessary to design fairness measures which are as robust as possible.
> We compare popular FR fairness measures, used by the NIST, depending on their uncertainty in Fig. 5 and conclude that the max-geomean metric is the more adequate. This result is supported by similar findings in the supplementary material B.2 (when varying the FR models, the sensitive attribute and the evaluation dataset). Note that this uncertainty quantification will be useful when evaluating the soundness of future fairness metrics.

---

> ### Author Response · Authors · 2023-11-21
> **Answer 2/4**
>
> Lastly, we provide in the supplementary material B.1 the coverage of the bands obtained with the recentered bootstrap. This coverage metric is the gold standard to quantify the accuracy of any method which builds confidence bands. The ideal property of a confidence interval is that specifying its confidence level $1-\alpha_{CI}$ (also called the *nominal coverage*) should lead to an interval having truly a probability equal to $1-\alpha_{CI}$ to contain the true (unknown) quantity $\mathrm{ROC}(\alpha)$. The coverage measures what this probability truly is and it should be equal (or very close) to the specified nominal coverage (see B.1). From Fig. 6 and Table 1, we observe that, even with finite datasets, it is the case for the recentered bootstrap, which suggests that the latter provides an appropriate tool for assessing the uncertainty of the performance measure.
>
> [1] Huang, Yuge, et al. "Curricularface: Adaptive curriculum learning loss for deep face recognition." Proceedings of the IEEE/CVF conference on computer vision and pattern recognition. 2020.
>
> [2] Meng, Qiang, et al. "Magface: A universal representation for face recognition and quality assessment." Proceedings of the IEEE/CVF conference on computer vision and pattern recognition. 2021.
>
> [3] Boutros, Fadi, et al. "Elasticface: Elastic margin loss for deep face recognition." Proceedings of the IEEE/CVF conference on computer vision and pattern recognition. 2022.
>
> [4] Kim, Minchul, Anil K. Jain, and Xiaoming Liu. "Adaface: Quality adaptive margin for face recognition." Proceedings of the IEEE/CVF conference on computer vision and pattern recognition. 2022.
>
> [5] Grother, Patrick. "Face Recognition Vendor Test (FRVT) Part 8: Summarizing Demographic Differentials." (2022).
>
>
>
>
>
>
> 2.2. There is a lack of experimental evidence that the proposed uncertainty is a better evaluation metric to compare the performance and fairness of Face Recognition models.
>
> Our proposed uncertainty metric (Eq. 11) is not a substitute for ROC or fairness metrics. It is introduced for Fig. 5, as detailed above. More generally, this metric allows one to (i) compare the uncertainty at several FAR levels for a fixed model, (ii) compare the uncertainty of different models at any FAR level, and (iii) compare the uncertainty of different fairness metrics at any FAR level (Fig. 5). We only presented (iii) within the paper. This allows for a selection between existing/future fairness metrics depending on their robustness with respect to the randomness of the evaluation data at disposal.
> In the same spirit, our confidence bands give additional information about the ROC curve (or fairness metrics) without being a substitute for those metrics.
>
> We point out that any alternative uncertainty measure for the performance will be a by-product of the distribution of the empirical ROC curve. Regarding the uncertainty of (scalar) statistics measuring fairness, we agree that there is currently no full consensus about fairness metrics in FR but, as all of them will be necessarily summaries of the empirical ROC curves, the methodology promoted here can be applied to quantify the uncertainty for any fairness criterion.

---

> ### Author Response · Authors · 2023-11-21
> **Answer 3/4**
>
> 2.3. Besides, there is no comparison of the proposed method with other methodologies.
>
> This is a good remark. In the following, we explain how to compare confidence band methods and then we detail the chosen baselines.
>
> As said in Part 2 of our answer, the coverage metric is the gold standard to quantify the accuracy of any method which builds confidence bands. The ideal property of a confidence interval is that specifying its confidence level $1-\alpha_{CI}$ (also called the *nominal coverage*) should lead to an interval having truly a probability equal to $1-\alpha_{CI}$ to contain the true (unknown) quantity $\mathrm{ROC}(\alpha)$. The coverage measures what this probability truly is and it should be equal (or very close) to the specified nominal coverage (see B.1). This coverage metric is the only appropriate way of comparing our method to others. Among several methods for building confidence bands, the most accurate is the one having an estimated coverage which is the closest to the specified nominal coverage.
>
> We added two baselines for the revised version of the paper.
>
> The first one is the naive bootstrap of Bertail et al. (2008). The non pairwise setup of the latter work significantly underestimates the ROC curve, leading to strongly inaccurate confidence intervals, as seen in Fig. 9 and Table 2.
>
> As pointed out in the related works section of the paper, no method to quantify the uncertainty in measuring performance/fairness in FR is documented in the literature to the best of our knowledge. The one described and analyzed in this paper is the first one. As explained in Section 3.1, using the (Gaussian) asymptotic law of the empirical similarity ROC curve is not an option given the very complex form of its covariance structure (involving probability density functions). \
> That being said, we designed a more meaningful baseline than the naive bootstrap of Bertail et al. (2008). The sole reasonable alternative is to use the bootstrap technique we propose to evaluate the variance of the empirical ROC at a certain FAR level and build a Gaussian confidence interval based on it (see supplementary material B.1). As now shown in the revised version (see Fig. 9 and Table 2), the coverage of this Gaussian approximation is much more accurate than the naive bootstrap, but less satisfactory than our recentered bootstrap.

---

> ### Author Response · Authors · 2023-11-21
> **Answer 4/4**
>
> 1. This paper is not the first to apply the bootstrap method to ROC curves and there isn’t enough difference between it and Bertail’s work proposed in 2008. The concept of uncertainty proposed in this paper is relatively new in the field of ROC curve estimation, but it is too simplistic in the field of uncertainty estimation, which needs deeper elaboration.
>
> Indeed, this paper is not the first to use the bootstrap to assess the uncertainty of a ROC curve, but it is the first to propose a proven valid bootstrap method applying to similarity ROC curves, and to highlight bias issues related to the distribution of empirical ROC’s. The work of Bertail et al. (2008), specialized to i.i.d. data, does not apply to the pairwise setup, to the similarity scoring framework (e.g. Face Recognition) in particular.
>
> In Face Recognition, the evaluation step of any algorithm consists in forming all pairs of data and computing a score for each pair. This pairwise nature leads the empirical evaluation metrics (FAR, FRR) to be of the form of U-statistics, instead of being simple i.i.d. averages as in the work of Bertail et al. (2008). A direct consequence is that the scores averaged in the FAR/FRR metrics exhibit a complex dependence structure (see Section 3.1). This is not the case in the framework of Bertail et al. (2008) where the scores are independent.
>
> Besides, applying Bertail et al. (2008) to the pairwise setup of Face Recognition (the naive bootstrap) yields strongly inaccurate confidence bands, as highlighted in Figures 2-3. Because of the pairwise nature of the evaluation metrics, such a bootstrap procedure tends to underestimate a lot the FRR metric, resulting in confidence bands for the ROC curve which do not even contain the empirical ROC curve most of the time (see Section 3.2 and Fig. 3). The recentered bootstrap we present within the paper counteracts this strong underestimation.
>
> In addition, we show in the supplementary material B.1 the poor coverage of the naive bootstrap of Bertail et al. (2008), compared to the coverage of the recentered bootstrap (see Fig. 9 and Table 2). Thus, the naive bootstrap is not appropriate to measure the uncertainty of similarity ROC curves (it is appropriate for the non pairwise setup) while our recentered bootstrap has a coverage very close to the specified nominal coverage.
>
> Lastly, our work proves the validity of the recentered bootstrap for the ROC curve, but also for popular FR fairness metrics, which is not the case for the bootstrap of Bertail et. al (2008).
>
> The difference between our work and the method of Bertail et al. (2008) is hopefully clarified in the revised version of the paper (Related Works, Section 3.1, Section 3.2 and supplementary material B.1).
>
> Regarding the relevance of the approach to uncertainty in the performance of FR systems we consider, we underline that, as it is inherent in the finiteness of the sizes of evaluation datasets only, no concept could be more exhaustive that the distributions themselves of empirical similarity ROC curves, the gold standard in evaluation, and these distributions are precisely what the proposed methodology estimates with guarantees. \
> We provide lower/upper bounds of a confidence interval, around the empirical ROC curve, at any confidence level. Our Theorem 1 proves that the obtained interval asymptotically contains the true ROC curve with a probability equal to the specified confidence level, which is exactly what a confidence interval should do. By construction, the width of the confidence interval is directly related to the variance of the empirical ROC curve which one computes from data (the randomness coming from the data at disposal, i.e. the aleatoric uncertainty). The same asymptotic property is also proved for fairness metrics. Lastly, our method holds even in the non asymptotic regime, as detailed in B.1 with the coverage being very close to the nominal coverage (specified confidence level).

---

### Official Review · Reviewer_DZ4X · 2023-10-30

**Soundness:** 4 excellent
**Presentation:** 3 good
**Contribution:** 3 good
**Rating:** 8
**Confidence:** 3

**Summary:**

This paper introduces introduces a method to quantify the uncertainty of similarity ROC curves in order to make meaningful comparison between different FR models. A naive bootstrap procedure tends to underestimate the ROC curve. Thus, a recentering technique is proposed so that a scalar uncertainty measure for ROC and fairness metrics is defined. With the statistical analysis and numerical experiments, a discussion about the practical use of the uncertainty value is done to achieve more reliable decisions related to accuracy and fairness.

**Strengths:**

+ The problem to measure the uncertainty in FR similarity function is well motivated. While ROC curve has become the standard to compare face recognition models, inconsistent comparison might occur due to the differences in evaluation dataset. The proposed method is able to build the confidence bands around the performance/fairness metrics by incorporating the uncertainty measures.

+ The recentering bootstrap technique refines the naive bootstrap that tends to underestimate the empirical FRR as shown in Figure 3.

+ The numerical experiments in face recognition problem is done to discuss how the proposed uncertainty metric and confidence band around ROC curve can be used in real-application. If the upper bound of the confidence band is lower, then the empirical fairness is better.  The differences between upper bound and lower bound of the confidence means that the uncertainty is high, thus it is better to choose a method that has small difference especially in the case where a strict fairness constraint is needed.

+ Multiple fairness metric has been evaluated using multiple FR methods. It is shown the max geomean metric has the lowest uncertainty in terms of FAR and FRR.

**Weaknesses:**

- There is a lack of justification of why Adacos has lower uncertainty than Arcrface in fairness metric. To fairly compare the performance of Adacos and Arcface, training those methods on Fair Face Recognition dataset [A] might help to justify the performance better. Note that there are multiple FR methods [B,C] that focus on solving fairness problem in face recognition. Comparison with those methods might be useful to justify the proposed metric.

- Instead of the background and preliminaries, it is more important to include more comparison of fairness metrics with various FR methods in the supplementary material in the main manuscript. As the paper focuses on the application of fair recognition technology, it is important to add more justification on the corresponding problem.

Additional references
- [A] FairFace Challenge at ECCV 2020: Analyzing Bias in Face Recognition, ECCVW 2020
- [B] Consistent Instance False Positive Improves Fairness in Face Recognition, CVPR 2021
- [C] RamFace: Race Adaptive Margin Based Face Recognition for Racial Bias Mitigation, IJCB 2021

**Questions:**

* In Figure 4, the $FRR^{max}_{min}$ is used as the fairness metric, why don't the authors use the best fairness metrics in Figure 5 (max-geomean)? It could lead to more consistent analysis.
* Why do the authors not evaluate the method in the fair face recognition dataset?
* Does the training dataset affect the uncertainty of the face recognition methods?

---

> ### Author Response · Authors · 2023-11-23
> **Answer 1/2**
>
> We would like to thank you for your careful reading of the manuscript and your positive feedback. Here are our answers to your concerns.
>
> 1. There is a lack of justification of why Adacos has lower uncertainty than Arcface in fairness metric. To fairly compare the performance of Adacos and Arcface, training those methods on Fair Face Recognition dataset [A] might help to justify the performance better.
>
> Indeed, AdaCos exhibits a surprisingly low uncertainty of its FRR max-min fairness, compared to ArcFace. We did not find any valid reason for this. The only valid justification would be to make the link between their loss function and the uncertainty (or at least a proxy) of their ROC for each gender. This task is complex and would be a work/paper on its own. For instance, the AdaCos/ArcFace loss function does not use any gender label for training. So, one would be forced to measure an uncertainty proxy during training for each gender (e.g. the variance of the logits for each gender, i.e. of the similarities between the embedding and the prototype of its identity for each gender for instance). Even doing so, there is a difference between the uncertainty one could measure during training and the uncertainty we tackle within the paper, which is about the randomness of evaluation data for each identity. To make a simple analogy, in the same way as ArcFace may have a better empirical fairness than AdaCos at some FAR levels, one model could have better uncertainty than others. In both cases, those models are not a priori trained to have the best fairness/uncertainty. Note that, as seen in B.3, the observed lower uncertainty of FRR fairness for AdaCos (compared to ArcFace) holds when changing the fairness metric.
>
> In our opinion, training AdaCos/ArcFace on the Fair Face Recognition dataset would not change the conclusions. As the training set we consider (the MS-Celeb-1M-v1c-r dataset), the Fair Face Recognition training set is strongly imbalanced in terms of gender, skintone or age. Any model trained on those training sets, with a normal procedure, would be biased.
>
> 2. Does the training dataset affect the uncertainty of the face recognition methods?
>
> An interesting question would be to find what happens to the fairness/uncertainty of a model when trained on a dataset with varying proportions of attributes (e.g. gender). Here again, this would be a work on its own, as the paper [1] details the influence of the gender balance in training data on the resulting gender-driven ROC’s. This is the equivalent of measuring the impact of the gender balance in training data on the empirical fairness metrics. A similar work could extend the study with our work, adding the uncertainty. We would like to underline that the purpose of the paper is not about the influence of the training set on the uncertainty, but rather about means of measuring the uncertainty about the randomness of evaluation data for any model, allowing for more trustworthy comparisons.
>
> [1] Albiero, Vítor, Kai Zhang, and Kevin W. Bowyer. "How does gender balance in training data affect face recognition accuracy?." 2020 ieee international joint conference on biometrics (ijcb). IEEE, 2020.
>
> 3. Instead of the background and preliminaries, it is more important to include more comparison of fairness metrics with various FR methods in the supplementary material in the main manuscript. As the paper focuses on the application of fair recognition technology, it is important to add more justification on the corresponding problem.
>
> We think that the preliminaries are necessary to introduce our multi-sample statistical setup, as well as the performance/fairness metrics. Those are the ingredients which make the U-statistic nature of the empirical FAR/FRR clear, i.e. the main reason why the naive bootstrap fails and how to counteract the underestimation of the ROC. The paper has applications in fairness, but also in performance (ROC) as seen in Fig.1. In the revised version of the paper, we added a comparison of the fairness metrics for ArcFace in B.4. Note that we prove the consistency of our confidence intervals for the ROC curve and any derivative/by-product of the attribute-driven ROC’s (which can be fairness metrics or completely different purpose metrics). This means that a simple algorithm exists to compute the uncertainty of lots of metrics deriving from ROC curves.

---

> > ### Author Response · Authors · 2023-11-23
> > **Answer 2/2**
> >
> > 4. Why do the authors not evaluate the method in the fair face recognition dataset?
> >
> > We chose the MORPH dataset as it is widely used in FR fairness research. It is provided with gender and age labels, which we used as the sensitive attribute (e.g. in B.2).  We also used the RFW dataset in order to inspect the race label for the ArcFace model. It is now in the revised version of the paper (see B.4). This allows us to compare the behaviour of each fairness metric, as well as their confidence bands. We agree that we could have used the Fair Face Recognition dataset.
> >
> > 5. In Figure 4, the FRR max min fairness is used as the fairness metric, why don't the authors use the best fairness metrics in Figure 5 (max-geomean)? It could lead to more consistent analysis.
> >
> > We chose the FRR max-min fairness as it is the only one defined in the main text of the paper. The other fairness metrics are defined in the supplementary material, for conciseness.
> >
> > In the revised version of the paper, we added the section B.3 where we provide the equivalent of Figure 4 for all FRR fairness metrics. Note that the conclusions from Figure 4 are unchanged when considering other fairness metrics. In particular, the case of the FRR max-min fairness is very similar to the case of the max-geomean fairness. It seems that the only difference is the scale of the metric, which is not so surprising as both fairness metrics have very similar definitions.
> >
> > In fact, one conclusion from this new section B.3 is that the FRR fairness metrics of AdaCos and ArcFace all exhibit exactly the same behaviour. This striking fact highlights the fact that all fairness metrics measure the same performance differentials. However, one significant difference between those metrics is their uncertainty: the width of the confidence bands can be high, relatively to the value of the empirical fairness, making the fairness metric not so robust. The comparison of the uncertainty between fairness metrics is displayed in Figure 5 and in Section B.2. One of our main contributions is the conclusion that the max-geomean fairness is the more robust metric, allowing for more trustworthy decisions.
> >
> > 6. Additional experiments
> >
> > Finally, we added, in the revised version, more details about the coverage of our recentered bootstrap, as well as the coverage of some baseline methods (see B.1).

---

### Meta-Review · Area_Chair_oYKh · 2023-12-07

**Metareview:**

The paper investigates uncertainty assessment in estimating ROC curves using evaluation datasets, in the context of similarity scoring. The paper  shows the consistency of empirical similarity ROC curves and proposes a variant of the bootstrap approach developed in prior work to build confidence bands around performance/fairness metrics, in order to quantify their variability.

The paper is a timely piece of work given the importance of ROC curves in evaluating performance and fairness in various applications, including face recognition. The authors provided a detailed rebuttal that helped to address some key questions from the reviewers. In the end, one reviewer raised the score and another reviewer maintained the positive rating. The third reviewer kept the "marginally below the acceptance threshold" rating but raised some constructive questions, some of which have been addressed in the authors' rebuttal. The authors are encouraged to incorporate these key points into the final version of the paper.

**Justification For Why Not Higher Score:**

Apply the bootstrap method to ROC curves has been explored before and the paper could better highlight the key differences. A more comprehensive evaluation could also further strengthen the paper.

**Justification For Why Not Lower Score:**

Uncertainty assessment of ROC curves could benefit many potential applications.

---

### Decision · Program_Chairs · 2024-01-16

Accept (poster)